# ADAPTIVE OPTIMIZATION IN THE ∞-WIDTH LIMIT

**Etai Littwin**
Apple
elittwin@apple.com

**Greg Yang** *
Microsoft Research
gregyang@microsoft.com

## ABSTRACT

Recent works have developed detailed understanding of large neural networks' behaviors via their infinite-width limits, e.g., the neural tangent kernel (NTK) and the feature learning ($\mu$) limits. These theories were developed for stochastic gradient descent. Yet, in practice, all large NN are trained using Adam or other adaptive gradient optimizers (AGO), which are not covered by such previous works. Here, we close this gap via the Tensor Programs framework. Specifically, for deep MLPs, we derive the NTK and $\mu$ parametrizations as well as their infinite-width limits. We find 1) The NTK limit of AGO, in contrast to that of SGD, now depends nonlinearly on the loss derivative but nevertheless still fails to learn features; 2) this is fixed by the $\mu$ limit of AGO (as in the case of SGD). To obtain these results, we extend the Tensor Programs language with a new instruction that allows one to express the gradient processing done by AGOs.

## 1 INTRODUCTION

Infinite width limits of neural networks have been a major focus of study in the last several years, underlying some of the most profound recent breakthroughs in our theoretical understanding of deep learning. Specifically, two types of limits have garnered the lions share of attention from the research community. The kernel limit, popularized by the seminal work of Jacot et al. (2018) refers to a regime of training where weights remain roughly in their initialized values, and training may be entirely characterized in function space by a constant kernel of a particular form, which depends on the network architecture. While easier to analyze, this limit does not permit updates to the internal representation of the network, hence it cannot account for data dependent feature learning, a staple of deep learning in practice. In contrast, the $\mu$ limit (of which the well-known mean field limit is a specific case in 1-hidden-layer perceptrons) refers to a regime of training where the weights adapt to the data during training in a nonlinear fashion, facilitating representation learning. It was recently shown in Yang & Hu (2020) that, under vanilla gradient based training, the precise setting of various hyperparameters relating to initialization scale and learning rate determine the type of infinite-width limit one can associate with a trained neural network. Notably, the $\mu$ parameterization was identified as the unique parameterization which gives rise to "maximal" feature learning dynamics in the infinite-width limit, where maximal refers to the fact that every layer learns features. However, quite remarkably, no such limits have yet been formally established for adaptive gradient based optimization of neural networks, which we make the focus of the present paper. Our main results in the paper are the identification and prescription of two types of infinite-width limits relating to popular AGO, the counterparts of the kernel and feature learning limits for vanilla GD. For the kernel limit counterpart, we uncover a fundamentally different dynamics for adaptive optimization, referred to as the adaptive neural tangent kernel (ANTK) regime. In this limit, the training dynamics can no longer be described by kernel gradient descent, since the kernel function itself depends non-linearly on the loss derivative. Our results lay a clear path to theoretically analyze the implicit biases of AGO in the infinite-width limit.

**Key Technical Contribution:** Analyzing the dynamics of adaptive optimization of arbitrary neural network architectures in the infinite-width limit presents a major technical challenge. As a main technical tool, we build upon the TP framework introduced and developed in a series of recent papers Yang (2019; 2020a;b). At a high level, the mechanics of the TP technique involves 1) write

---

*Please see arxiv.org for the full, updated version of this paper

down the relevant neural network computation (e.g. the first forward pass in the NNGP case) as a principled composition of matrix multiplication and coordinatewise nonlinearities, called a Tensor Program, and 2) recursively calculate the distribution of coordinates of each vector via what's called the Master Theorem. However flexible, the "language" of TP is not expressive enough to represent the necessary computations involving adaptive optimization since it does not support the application of nonlinear functions to high order tensors. In the present paper, we solve this issue by expanding the TP framework with additional functionities, and proving a new master theorem which enables our analysis. While we present a simple application of our new framework on MLPs in Theorem 4.1 and Theorem 4.2, it is applicable in a much wider setting, including most practical architectures and algorithms. As an additional technical contribution, we prove a $O(n^{-1/2})$ (where $n$ represents the width) convergence rate guarantee for all variables produced by the program, which might be of independent interest.

**Our Contributions:** This paper presents the following major contributions:

1. We present the first rigorous infinite-width analysis of adaptive optimization of MLPs parameterized using the ANTK and $\mu$ parameterizations. Our results rigorously equate training of such networks to discrete time dynamical equations.

2. We develop a new tensor program framework along convergence rate guarantees, unlocking the infinite-width analysis of adaptive optimization in an architecturally universal sense.

**Paper Organization:** This paper is organized as follows: We survey related work in Section 2. In Section 3 we set up preliminaries and notations used extensively in Section 4. In Section 4 we illustrate ANTK and $\mu$ limits for MLPs. Section 5 is dedicated to a formal introduction to the new TP framework. Although it is used as a main tool to prove our results in Section 4, Section 5 is more general and can be read as a standalone.

## 2 RELATED WORK

A large body of literature exists on both the kernel (NTK) limit Arora et al. (2019); Jacot et al. (2018); Lee et al. (2019); Yang (2020c); Yang & Littwin (2021) and the mean field limit for 2 layer neural network Chizat & Bach (2018); Mei et al. (2018b); Nguyen & Pham (2020); Rotskoff & Vanden-Eijnden (2018); Sirignano & Spiliopoulos (2020). Various papers describe the kernel and feature learning regimes more generally without taking an infinite-width limit. Chizat et al. (2019) describes the "lazy training" regime in arbitrary differentiable programs, and is controlled by a single parameter $\alpha$ which scales the output. It is shown that when $\alpha$ is large, the weight need only move slightly to fit the training data, and network essentially performs kernel learning. Many papers Allen-Zhu et al. (2019); Huang & Yau (2020); Mei et al. (2018a) view the kernel and feature learning regimes as learning in different timescales, explicitly incorporating the time dependence in the infinite-width limit, and others derive finite width corrections to the NTK for finite width networks Hanin & Nica (2020); Littwin et al. (2020a). In this paper, we consider training time to be constant, and take only the width to infinity. This way, kernel and feature learning behaviour are separated by the parameterization employed at initialization, and not width or training time. TPs, first introduced in Yang (2019) and expanded upon in Yang (2020a;b), were developed as a theoretical framework to analyze the infinite-width limits of any architecture expressible in the TP language, in an attempt to rid the per architecture analysis prevalent in the literature Alemohammad et al. (2021); Du et al. (2019); Hron et al. (2020); Littwin et al. (2020b). Yang & Hu (2020) defined a natural space of neural network parametrizations (abc-parametrizations), and classified all resulting infinite-width limits into two possible catagories: 1) the kernel limit, in which weights and activations remain roughly in their initialization state, and 2) feature learning limit, in which weights move substantially and adapt to data. The $\mu$ parameterization was then identified as the "optimal" parameterization for arbitrary architectures in which all layers learn features, and was later heuristically extended to AGOs Yang et al. (2021). Unrelated, AGOs Duchi et al. (2010); Kingma & Ba (2015); Zhou et al. (2018) and their variants were developed to accelerate learning by adapting the learning rate on a per parameter basis, and currently serve as a prerequisite for training large scale transformer models Huang et al. (2020); Liu et al. (2020); Zhang et al. (2019). Crucially, no previous work has yet developed a theory for infinite-width neural network trained with AGOs.

## 3 PRELIMINARIES

**Adaptive Optimizers:** Generically, if $g_0, g_1, ..., g_t \in \mathbb{R}$ denote the gradients of some scalar parameter $w \in \mathbb{R}$ at steps $0, 1, ..., t$, an adaptive update $\Delta w_t = w_{t+1} - w_t$ at step $t$ takes the form $\Delta w_t = -\eta \frac{m}{\sqrt{v}+\epsilon}$ where $\eta$ is the learning rate and $m$ and $v$ are both functions of the past gradients $g_0, \ldots, g_t$. For example, in Adam, $m$ and $v$ are the exponential moving averages of $g_{(i)}$ and $g_{(i)}^2$. Here, we consider an even more general notion of adaptive updates, encompassing all modern AGOs.

**Definition 3.1.** We say an update $\Delta w_t \propto Q_t(g_0, g_1, ..., g_t; \epsilon)$ to a weight $w$ at time $t$ is *adaptive* if it is proportional (up to a constant factor) to a function $Q_t : \mathbb{R}^{t+1} \to \mathbb{R}$ such that $\forall_{c \neq 0},\ Q_t(cg_0, cg_1, ..., cg_t; c\epsilon) = Q_t(g_0, g_1, ..., g_t; \epsilon)$. Moreover, if $Q_t(g_0, g_2, ..., g_t; \epsilon) = Q(g_t; \epsilon)$ (only depends on $g_t$), then we furthermore say $\Delta w_t$ is *memoryless*.

To maximize clarity, we focus on the simpler case of memoryless adaptive updates in the main text. For example, in Adam this implies setting $\beta_1, \beta_2 = 0$. This simplification will already highlight the key differences between the adaptive and non-adaptive case. We provide an extension of these results to the case of AGOs with memory in Appendix C, and provide numerical verification of our results in Appendix D.

**MLPs and ABC(D) Parameterization:** We use a standard scalar output MLP $f$ with $L$ hidden layers as a working example to illustrate the adaptive kernel and feature learning limits. Given an input sample $\xi \in \mathbb{R}^{d_{\text{in}}}$, weight matrices $W^{L+1} \in \mathbb{R}^n, \{W^l\}_{l=2}^L \in \mathbb{R}^{n \times n}, W^1 \in \mathbb{R}^{n \times d_{\text{in}}}$ and an activation function $\phi$ which we assume has a pseudo lipschitz first derivative, the output $f(\xi) \in \mathbb{R}$ is given by:

$$f(\xi) = W^{L+1\top} x^L(\xi), \quad \begin{cases} x^l(\xi) = \phi(h^l(\xi)) & \text{for } 1 \leq l \leq L \\ x^0(\xi) = \xi \end{cases}, \quad \forall_{1 \leq l \leq L}\ h^l(\xi) = W^l x^{l-1}(\xi) \quad (1)$$

We adopt the ABC parameterization convention from Yang & Hu (2020). Namely, for any layer $l$, each weight matrix is parameterized using $W = n^{-a_l} w^l$ where $w^l$ are the learnable weights, which are initially sampled iid from a normal distribution $\mathcal{N}(0, n^{-2b_l})$. Finally, the learning rate is parameterized using $\eta n^{-c_l}$ where we plug $\eta = 1$ for simplicity. In this paper, we assign specific values to $\{a_l\}_l, \{b_l\}_l, \{c_l\}_l$ for the ANTK and $\mu$ parameterizations. Additionally, we will parameterize the $\epsilon$ parameter in the AGO with $\epsilon_l = n^{-d_l} \epsilon$, where $\epsilon > 0$. The per-layer scaling for $\epsilon_l$ will turn out to be crucial to prevent the adaptive gradients from collapsing to either 0 or a step function as $n \to \infty$. We summarize the two parameterizations in the following table:

| Parameterization | $a_l$ | $b_l$ | $c_l$ | $d_l$ |
|---|---|---|---|---|
| ANTK | $\begin{cases} \frac{1}{2} & l > 1 \\ 0 & \text{else} \end{cases}$ | $0$ | $\begin{cases} 1 & L+1 > l > 1 \\ \frac{1}{2} & \text{else} \end{cases}$ | $\begin{cases} 1 & L+1 > l > 1 \\ \frac{1}{2} & \text{else} \end{cases}$ |
| $\mu$ | $\begin{cases} -\frac{1}{2} & l = 1 \\ \frac{1}{2} & l = L+1 \\ 0 & \text{else} \end{cases}$ | $\frac{1}{2}$ | $\begin{cases} 1 & L+1 > l > 1 \\ \frac{1}{2} & \text{else} \end{cases}$ | $\begin{cases} 1 & L+1 > l > 1 \\ \frac{1}{2} & \text{else} \end{cases}$ |

Table 1: ANTK and $\mu$ parameterizations.

**Representing (pre)Activation Vectors via Random Variables:** As we will see, as width becomes large, the entries of the activation and preactivation vectors will become roughly iid (just like in the SGD case), both at initialization (which is easy to see) and training (which is harder to see). Hence a vector's behavior can be tracked via a random variable that reflects the distribution of its entries. Concretely, if $x \in \mathbb{R}^n$ is one such vector, then we write $Z^x$ for such a random variable, such that $x$'s entries look like iid samples from $Z^x$. When $x$ is scaled to have typical entry size independent of $n$,[1] then $Z^x$ can be taken to be a random variable independent of $n$ as well. In general, given two such vectors $x, y \in \mathbb{R}^n$, their random variables $Z^x$ and $Z^y$ will be correlated, in such a

---

[1] i.e., $\|x\|^2 / n = \Theta(1)$ as $n \to \infty$

way that $\lim_{n\to\infty} \frac{x^\top y}{n} = \mathbb{E}Z^x Z^y$. Generally, inferring with initialized networks entail computing expectations with gaussian $Z$ variables, which take a relatively simple form. However, a fundamental question is how the $Z$ variables evolve during training, which we address next.

# 4 ADAPTIVE OPTIMIZATION OF AN MLP

In the following section we illustrate the infinite-width limits of adaptive optimization for simple MLPs. For each parameterization, we begin by laying the basic intuition, culminating in Theorem 4.1 and Theorem 4.2. For a cleaner presentation, we assume the first and last layers are fixed, however our results are easily extended to the general case. In our setup we assume the network is trained using an AGO according to Definition 3.1, and a batchsize of 1.

**Notations:** Slightly abusing notation, we use subscripts to denote both the step index $t$, and coordinates of vectors with $\alpha, \beta$. We assume $\xi_t$ is a training sample fed to the neural network at step $t$ (starting from $\xi_0$). and we use $y_t(\xi)$ for any input dependent vector/scalar $y$ to denote its evaluation given $\xi$ at step $t$. To reduce clutter we remove explicit dependency on input if it is implied by the step index (i.e $y_t = y_t(\xi_t)$ and $y = y_0(\xi_0)$). We use $\tilde{y}_t = y_t(\tilde{\xi})$ to express the dependency of $y$ on an arbitrary input $\tilde{\xi}$ at step $t$. We will also denote $\Delta y_t(\xi) = y_{t+1}(\xi) - y_t(\xi)$ and $\delta y_t(\xi) = \sqrt{n}\Delta y_t(\xi)$. We assume the network is trained using a generic loss function $\mathcal{L}$, with a loss derivative $\mathcal{L}'_t = \nabla_{f_t}\mathcal{L}_t$. We use the notation $dh^l$ **based on context**: for ANTK parameterization, we set $dh^l(\xi) \stackrel{\text{def}}{=} \sqrt{n}\frac{\partial f}{\partial h^l}(\xi) \in \mathbb{R}^n$, whereas for $\mu$ parameterization, we set $dh^l(\xi) \stackrel{\text{def}}{=} n\frac{\partial f}{\partial h^l}(\xi) \in \mathbb{R}^n$. This context dependent notation is convenient since it insures that the components of $dh^l(\xi)$ are roughly in the order of $\Theta(1)$ for both parameterizations. Finally, we use $\mathring{\bullet}$ to denote the infinite-width limit of a (possibly random) scalar $\bullet$ (i.e $\lim_{n\to\infty}\tilde{f}_t = \mathring{\tilde{f}}_t$). Using the above notation, we can express the gradient of any intermediate layer $w^l$ at step $t$ for both parameterizations by $\frac{1}{n}dh^l_t x^{l-1\top}_t \mathcal{L}_t$. Using Definition 3.1 the adaptive weight update $\Delta w^l$ for both parameterizations is given by:

$$\forall_{1 < l \le L}, \quad \Delta w^l_t = -\frac{1}{n}Q(\frac{1}{n}dh^l_t x^{l-1\top}_t \mathcal{L}'_t; \frac{\epsilon}{n}) = -\frac{1}{n}Q(dh^l_t x^{l-1\top}_t \mathcal{L}'_t; \epsilon) \tag{2}$$

where the function $Q$ is applied element-wise on the matrix $dh^l_t x^{l-1\top}_t \mathcal{L}'_t \in \mathbb{R}^{n \times n}$. For the remainder of the paper we will suppress the explicit the dependency on $\epsilon$ and simply absorb it into $Q$.

## 4.1 THE ANTK LIMIT

In the NTK limit, intuitively, the weights of the neural network move by a negligible amount during training, such that the network function may be approximated by its first order Taylor expansion around its initialization. This intuition carries over to the ANTK limit as well. At a high level, the following hold at any step $t$: $\Delta\tilde{h}^l_t$ for any layer $l$ will be of order $\Theta(n^{-\frac{1}{2}})$. By definition $\tilde{h}^l_{t+1} = \tilde{h}^l_t + \Delta\tilde{h}^l_t$, hence the coordinates of $\tilde{h}^l_t$ for any layer $l$ do not change in the limit, and, for any input, the coordinate distributions remain constant $\forall_{l\in[1,l]}Z^{\tilde{h}^l_t} = Z^{\tilde{h}^l}, Z^{d\tilde{h}^l_t} = Z^{d\tilde{h}^l}$. Instead of training $f$, we consider training the first order linearization of $f$, denoted by $f^{\text{lin}}$, around its initial parameters. The function updates $\Delta\tilde{f}^{\text{lin}}_t$ are given by

$$\Delta\tilde{f}^{\text{lin}}_t = -\frac{1}{n^2}\sum_{l=2}^{L} d\tilde{h}^{l\top} Q(dh^l_t x^{l-1}_t \mathcal{L}'_t)\tilde{x}^{l-1} \tag{3}$$

Under the ANTK parameterization, as with SGD, training $f^{\text{lin}}$ and $f$ using AGO is equivalent. The following theorem describes the evolution of $\mathring{\tilde{f}}_t$ exactly:

**Theorem 4.1.** *Let $f(\xi) \in \mathbb{R}$ denote an MLP as in Eq. (1) parameterized using the ANTK parameterization described in Section 4.1, where $\phi'$ is pseudo-Lipschitz. Assume layers $\{w^l\}_{l=2}^{L}$ are trained using a memoryless AGO with a pseudo-Lipschitz function $Q$ according to Definition 3.1 and a batchsize of 1, using a loss function $\mathcal{L}$ with a pseudo-Lipschitz first derivative. Then, at any step $t$*

and for any sample $\tilde{\xi}$, it holds that $\tilde{f}_t \xrightarrow{a.s} \mathring{\tilde{f}}_t$ where $\Delta \mathring{\tilde{f}}_t = -\mathcal{K}_{Adp}(\xi_t, \tilde{\xi} | \mathring{\mathcal{L}}_t')$, where:

$$\mathcal{K}_{Adp}(\xi_t, \tilde{\xi} | \mathring{\mathcal{L}}_t') = \sum_{l=2}^{L} \mathbb{E}\big[ Z^{d\tilde{h}^l} Q \big( Z^{dh^l(\xi_t)} Z^{x^{l-1}(\xi_t)} \mathring{\mathcal{L}}_t' \big) Z^{\tilde{x}^{l-1}} \big] \tag{4}$$

$$\mathring{\mathcal{L}}_t' = \mathcal{L}_t'(\mathring{\tilde{f}}_t(\xi_t)) \tag{5}$$

where the expectation is taken over all $Z$ variables at initialization.

Let us discuss Theorem 4.1 in a bit more detail. First, note that after the output values $\tilde{f}_t$, and by extension the loss derivatives $\mathcal{L}_t'$ are deterministic after conditioning on the outputs $f$ at initialization, hence the only source of randomness in Eq. (162) is from the $Z$ variables at initialization [2]. Second, it is straightforward to show that by setting $Q(x) = x$ we would get the SGD equivalent (when setting the learning rate to be 1) of Eq. (162), which takes the form $\tilde{f}_t^{\text{sgd}} \approx -\sum_{l=2}^{L} \frac{dh_t^{l\top} d\tilde{h}_t^l}{n} \frac{x_t^{l-1\top} \tilde{x}_t^{l-1}}{n} \mathcal{L}_t'$. For SGD, one may naively apply the the law of large numbers argument (LLN) and derive the infinite-width limit under plain SGD: $\Delta \mathring{\tilde{f}}_t^{\text{sgd}} \xrightarrow{a.s} -\mathcal{K}(\xi_t, \tilde{\xi}) \mathring{\mathcal{L}}_t'$ where $\mathcal{K}$ is the NTK function defined as:

$$\mathcal{K}(\xi_t, \tilde{\xi}) = \sum_{l=2}^{L} \mathbb{E}[Z^{dh^l(\xi_t)} Z^{d\tilde{h}^l}] \mathbb{E}[Z^{x^{l-1}(\xi_t)} Z^{\tilde{x}^{l-1}}] \tag{6}$$

Hence Theorem 4.1 is a generalization of the well known NTK limit. At a glance, the transition from Eq. (3) to its infinite-width counterpart seems like a straightforward application of LLN. However, the validity of Theorem 4.1 is not at all straightforward, and cannot be obtained by applying gaussian conditioning based arguments as in Yang & Littwin (2021), even for the first weight update. Technically, the complication arises from nonlinearity of the $Q$ function: unlike SGD where nonlinear functions are only applied to vectors, in Eq. (3) we construct a matrix (more generally a tensor) using a nonlinear function $Q$. Operations of this type make even the simplest case, where all inputs are iid gaussian, tricky to analyze. Developing a general framework to handle such operations will be key to developing a general framework to prove our main results later on. We discuss this technicality in more detail in Section 6.

For general adaptive updates, Theorem 4.1 implies that $\mathcal{K}_{\text{Adp}}$ is nonlinear in the loss derivative $\mathcal{L}_t'$, inducing a fundamentally different dynamics then the kernel gradient descent featured with SGD, and we leave the more in depth analysis of its nature for future work. Similar to NTK however, the ANTK limit does not allow data dependent feature learning, since the weights and activations remain roughly at their initialized values. This allows us to adopt a function space view of the training dynamics, where the output updates depend solely on the values of the outputs in the previous iteration, and without any dependence on the state of the internal representation computed by the network, which remain unchanged. In contrast, the $\mu$ parameterization allows data dependent feature learning in the infinite-width limit, which we analyze next.

## 4.2 Feature Learning with $\mu$ parameterization

We now turn to analyzing the infinite-width training dynamics under $\mu$ parameterization. Fundamentally, each weight update $\Delta w_t^l$ will cause each preactivation vector $\tilde{h}_t^l$ to change entrywise by something of order $\Theta(1)$, and the coordinate distributions at the limit will evolve non-trivially. Generally, the dynamical equations equivalent of an infinite-width neural network in the feature learning regime (using $\mu$ or otherwise) is much more complex to derive for deep networks. Although our new TP formulation discussed in Section 5 provides a complete set of tools to tackle most architectures, we will be content with illustrating the main points using a 2 hidden layer MLP where only the middle layer weights $w^2$ are trained. Using Eq. (2), we can express $\tilde{h}_{t+1}^2, \tilde{x}_{t+1}^2, \tilde{f}_{t+1}$ using:

$$\tilde{h}_{t+1}^2 = \tilde{h}_t^2 - \frac{1}{n} Q(dh_t^2 x_t^{1\top} \mathcal{L}_t') \tilde{x}^1, \quad \tilde{x}_{t+1}^2 = \phi(\tilde{h}_{t+1}^2), \quad \tilde{f}_{t+1} = \frac{\sqrt{n} w^{3\top} \tilde{x}_{(t+1)}^2}{n} \tag{7}$$

Note that under $\mu$ the coordinates of $\sqrt{n} w^3$ are randomly distributed according as $\mathcal{N}(0, 1)$, hence we expect $\Delta \tilde{f}_t$ to be $\Theta(1)$. Due to the $Q$ function applied to the gradient, the components of the

---

[2] The $Z$ variables are in fact independent from the outputs $f(\xi)$. This is made rigorous in the proof.

$\mathbb{R}^{n \times n}$ matrix $Q(dh_t^2 x_t \mathcal{L}_t')$ are not vanishing as $n \to \infty$, and the update $\frac{1}{n}Q(dh_t^2 x_t^{1\top}\mathcal{L}_t')\tilde{x}^1$ is by consequence generally not vanishing as well. Since the updates $\Delta\tilde{h}_t^2$ are non vanishing, the feature vector $\tilde{x}_t^2$ evolves substantially (for non degenerate $\phi$), which enables feature learning. Taking the limit of Eq. (7) to derive dynamical equations is again not an easy task. Consider the case where $Q = \text{Identity}$, which results in the update equation $\tilde{h}_{t+1}^2 = \tilde{h}_t^2 - \frac{\tilde{x}^{1\top}x^1(\xi_t)}{n}dh_t^2\mathcal{L}_t'$, and can be expressed purely using operations between vectors. For general nonlinear $Q$ functions, we must deal with a matrix - vector multiplication as in Eq. (7). This implies that unlike with SGD where we must reason about how a finite collection of $\mathbb{R}^n$ vectors behave in the limit, we must now reason about the behaviour of $\mathbb{R}^{n \times n}$ matrices (see Section 6 for further discussion on this). The following theorem describes the evolution of $\mathring{\tilde{f}}_t$ under $\mu$ exactly:

**Theorem 4.2.** *Let $f(\xi) \in \mathbb{R}$ denote an MLP as in Eq. (1) with $L = 2$ parameterized using the $\mu$ parameterization described in Section 4.1, where $\phi'$ is pseudo-Lipschitz. Assume layers $w^2$ is trained using an AGO with a pseudo-Lipschitz function $Q$ function according to Definition 3.1 and a batchsize of 1, using a loss function $\mathcal{L}$ with a pseudo-Lipschitz first derivative. Then at any step $t$ and for any sample $\tilde{\xi}$, it holds that $\tilde{f}_t \overset{a.s}{\to} \mathring{\tilde{f}}_t$ where $\mathring{\tilde{f}}_t$ can be computed as follows:*

$$Z^{\tilde{h}_{t+1}^2} = Z^{\tilde{h}_t^2} - \mathbb{E}_{Z^{x^1(\xi_t)}, Z^{\tilde{x}^1}}\left[Q(\zeta\phi'(Z^{h_t^2})Z^{x^1(\xi_t)}\mathring{\mathcal{L}}_t)Z^{\tilde{x}^1}\right] \tag{8}$$

$$Z^{\tilde{x}_t^2} = \phi(Z^{\tilde{h}_t^2}), \quad \mathring{f}_0 = 0, \quad \mathring{f}_t = \mathbb{E}[\zeta Z^{\tilde{x}_t^2}], \quad \mathring{\mathcal{L}}_t' = \mathcal{L}_t'(\mathring{f}_t(\xi_t)) \tag{9}$$

*where the expectations are taken over all $Z$ variables (including $\zeta \overset{\mathrm{d}}{=} \mathcal{N}(0,1)$).*[3]

From Theorem 4.2 it is clear in what sense $\mu$ parameterization enables feature learning in the limit: from Eq. (8) the random variable encoding the updates to the hidden representation $Z^{\tilde{h}_t^2}$ is of order $\Theta(1)$ for non degenerate $Q$ and $\phi$ functions, and is generally no longer gaussian at steps $t > 0$, allowing the neural network to learn data dependent features. Once again, substituting $Q(x) = x$ would default the equations in Eq. (8) back to those of SGD with an appropriate step size, hence our Theorem 4.2 generalizes feature learning with SGD.

## 5 A TENSOR PROGRAM FOR ADAPTIVE OPTIMIZERS

In Section 4 we have derived two types of limits in a relatively restricted setting of training an MLP. In the following section, we go into more detail into the TP framework that allows such principled derivations in a much broader setting. While doing so, we will highlight the additional functionities introduced in the present paper that are key to unlocking the analysis of adaptive optimization, and removing certain assumptions preventing previous iterations from achieving full architectural generality. In the previous section we have provided intuitive calculations for computing the infinite-width limits of trained neural networks by explicitly expressing the updates to the network internal representation and output at step $t$, and then naively converting coordinates of vectors to iid samples from a known distribution (the $Z$ variables) . However, these computations become exceedingly complex with an arbitrary number of updates and complex architectures, and it is not clear whether these arguments can be applied in a general setting. Tensor programs is a framework designed to automate these computations, while providing theoretical guarantees. Generally, computations involving adaptive optimization of most architectures contain a few repeating operations (i,e matrix multiplications, point wise non linearities...), applied to variables of different types (i.e matrices, vectors and scalars). This brings forth the notion of a program: A directed computational graph where nodes represent variables (typically $\mathbb{R}^n$ vectors or scalars), and edges represent operations performed on the variables. As a concrete example, the forward pass of an MLP given some input can be expressed as a tensor program where the input represents the root node, and the affine transformation in each layer represent an edge between nodes. We give a more formal description of our framework in the following.

### 5.1 NE⊗ORT PROGRAMS

**Definition 5.1.** A NE⊗ORT program is a sequence of $\mathbb{R}^n$-vectors and $\mathbb{R}$-scalars inductively generated via one of the following ways from an initial set $\mathcal{C}$ of random scalars, $\mathcal{V}$ of random $\mathbb{R}^n$ vectors, and

---

[3]Once again, the loss derivatives $\mathcal{L}_t'$ are deterministic in Eq. (8)

a set $\mathcal{W}$ of random $\mathbb{R}^{n \times n}$ matrices (which will be sampled with iid Gaussian entries in Setup 5.2). Concretely, using weights $\mathcal{W}$ and some pseudo-lipschitz function $\psi : \mathbb{R}^{k(r+1)+l} \to \mathbb{R}$ for $k, l, r \in \mathbb{N}$, the program generates new vectors and scalars from previously generated vectors $\boldsymbol{x} = \{x^1, ..., x^k\} \in \mathbb{R}^n$ and scalars $\Theta = \{\theta_1, \theta_2, ..., \theta_l\} \in \mathbb{R}$ by the following instructions (using the notation $\boldsymbol{x}_i = \{x_i^1, ..., x_i^k\}$):

TENSOR Generates a vector $x \in \mathbb{R}^n$ by $x_\alpha = \frac{1}{n^r} \sum_{\beta_1, ..., \beta_r = 1}^n \psi(\boldsymbol{x}_\alpha, \boldsymbol{x}_{\beta_1}, ..., \boldsymbol{x}_{\beta_r}; \Theta)$.

TENSORMOMENT Generates a scalar $\theta \in \mathbb{R}$ by $\theta = \frac{1}{n^{r+1}} \sum_{\alpha, \beta_1, ..., \beta_r = 1}^n \psi(\boldsymbol{x}_\alpha, \boldsymbol{x}_{\beta_1}, ..., \boldsymbol{x}_{\beta_r}; \Theta)$.

MATMUL Generates a vector $x \in \mathbb{R}^n$ by $x = W\bar{x}$ or $x = W^\top \bar{x}$ where $\bar{x} \in \boldsymbol{x}, W \in \mathcal{W}$.

Let us unpack Definition 5.1. We can think of the TENSOR operation as a generalized version of the standard pointwise nonlinearity which acts on vectors (or tensors where only one dimension increases to infinity, akin to the NONLIN instruction in Yang (2020b). Instead, the TENSOR instruction applies a pointwise nonlinearity $\psi$ to a tensor of rank $r + 1$ where all dimensions are of size $n$, and then contracts $r$ dimensions to produce a vector. We note that the instruction subsumes the standard applications of (non)linear functions applied to vectors by setting $r = 0$. The TENSORMOMENT operation allows us to fully contract a tensor of rank $r + 1$ to a scalar. Finally, the MATMUL operation is copied over from Yang (2020b), and implements a standard linear layer.

The initial sets of vectors and scalars $\mathcal{V}, \mathcal{C}$, and weights $\mathcal{W}$ are randomly sampled according to Setup 5.2:

**Setup 5.2.** *1) For each initial $W \in \mathcal{W}$, we sample iid $W_{\alpha\beta} \sim \mathcal{N}(0, \sigma_W^2/n)$ for some variance $\sigma_W^2$ associated to $W$, independent of other $W' \in \mathcal{W}$; 2) for some multivariate Gaussian $Z^\mathcal{V} = \{Z^x : x \in \mathcal{V}\} \in \mathbb{R}^\mathcal{V}$, we sample the initial set of vectors $\mathcal{V}$ like $\{x_\alpha : x \in \mathcal{V}\} \sim Z^\mathcal{V}$ iid for each $\alpha \in [n]$. 3) For each initial scalar $\theta \in \mathcal{C}$, we require $\forall_{p>0}, n^{1-p}(\theta - \mathring{\theta})^2 \xrightarrow{a.s.} 0$ for some deterministic $\mathring{\theta} \in \mathbb{R}$.*

Note that the initial set of vectors $\mathcal{V}$ are assumed to be gaussian in $\mathbb{R}^n$. In a typical neural network training scenario, the initial vectors correspond to input/output layer weights and biases at initialization, and the initial matrices correspond to hidden layer weights at initialization, so their Gaussian sampling reflects their Gaussian initialization.

**Example:** A program encoding the first forward, backward and adaptive update (Eq. (7)) using $\mu$ parameterization is provided in Table 2

| Expression | Op type | Implementation |
|---|---|---|
| $h^2 = W^2 x^1$ | MATMUL | |
| $x^2 = \phi(h^2)$ | TENSOR | $\psi(h^2)$ for $\psi(a) = \phi(a)$ |
| $f = \frac{\sqrt{n} w^{3\top} x^2}{n}$ | TENSORMOMENT | $\frac{1}{n} \sum_{\alpha=1}^n \psi(\sqrt{n} w_\alpha^3, x_\alpha^2)$ for $\psi(a, b) = ab$ |
| $\mathcal{L}'(f)$ | TENSORMOMENT | $\frac{1}{n} \sum_{\alpha=1}^n \psi(; f)$ for $\psi(; \theta) = \mathcal{L}'(\theta)$ |
| $dh^2$ | TENSOR | $\psi(\sqrt{n} w^3, h^2)$ for $\psi(a, b) = a\phi'(b)$ |
| $\Delta \tilde{h}^2$ | TENSOR | $\frac{1}{n} \sum_{\beta=1}^n \psi(dh^2, x_\beta^1, \tilde{x}_\beta^1; \mathcal{L}')$ for $\psi(a, b, c; \theta) = Q(ab\theta)c$ |

Table 2: A NE⊗ORT Program encoding the forward/backward and adaptive update of an MLP. In the above, $a, b, c, \theta \in \mathbb{R}$ represent inputs to some function $\psi$ implementing a TENSOR or a TENSORMOMENT instruction.

## 5.2 THE MASTER THEOREM

We can guarantee certain properties hold for vectors and scalars generated by a tensor program in the infinite-width limit. In short, any generated scalar $\theta$ will almost surely converge to a deterministic limit $\mathring{\theta}$ as $n \to \infty$, at a rate of $O(\frac{1}{\sqrt{n}})$. For any generated vector $x \in \mathbb{R}^n$, the coordinates of $x$ will approach iid samples from some distribution. Adopting the notation from Section 3, we denote by

$Z^x$ a random variable distributed like the coordinates of $x$ as $n \to \infty$. The following constructs the random variable $Z^x$ for every vector $x$ and a deterministic scalar $\mathring{\theta}$ for every scalar $\theta$ in the program, where we assume $\boldsymbol{x} = \{x^1, ..., x^k\}, \Theta = \{\theta_1, ..., \theta_l\}$ are previously generated vectors and scalars, and we use the abbreviated $Z^{\boldsymbol{x}}$ to denote the set of $k$ random variables $\{Z^{x^i}\}_{i=1}^k$ for all $x^i \in \boldsymbol{x}$.

**Definition 5.3** ($Z^x$ and $\mathring{\theta}$)**.** We recursively define $Z^x \stackrel{\text{def}}{=} \hat{Z}^x + \dot{Z}^x$ for each vector $x$ and $\mathring{\theta}$ for each scalar $\theta$ as follows:

**ZINIT** If $x \in \mathcal{V}$, then $Z^x$ is defined as in Setup 5.2. We also set $\hat{Z}^x \stackrel{\text{def}}{=} Z^x$ and $\dot{Z}^x \stackrel{\text{def}}{=} 0$.

**ZTENSOR** If $x$ is generated by TENSOR (see Definition 5.1), then $Z^x \stackrel{\text{def}}{=} f(Z^{\boldsymbol{x}})$ where $f(\zeta) \stackrel{\text{def}}{=} \mathbb{E}_{Z_1^{\boldsymbol{x}}, ..., Z_r^{\boldsymbol{x}}}[\psi(\zeta, Z_1^{\boldsymbol{x}}, Z_2^{\boldsymbol{x}}, Z_r^{\boldsymbol{x}}; \mathring{\Theta})]$ with $Z_i^{\boldsymbol{x}}$ being iid copies of $Z^{\boldsymbol{x}}$.

**ZTENSORMOMENT** If $\theta$ is generated by TENSORMOMENT (see Definition 5.1), then $\mathring{\theta} \stackrel{\text{def}}{=} \mathbb{E}_{Z^{\boldsymbol{x}}, Z_1^{\boldsymbol{x}}, ..., Z_r^{\boldsymbol{x}}}[\psi(Z^{\boldsymbol{x}}, Z_1^{\boldsymbol{x}}, ..., Z_r^{\boldsymbol{x}}; \mathring{\Theta})]$. Here $\mathring{\Theta} = \mathring{\theta}_1, ..., \mathring{\theta}_l$ are deterministic, so the expectation is taken over $Z^{\boldsymbol{x}}, Z_1^{\boldsymbol{x}}, ..., Z_r^{\boldsymbol{x}}$, where $\{Z_j^{\boldsymbol{x}}\}_{j=1}^r$ are $r$ iid samples drawn from the same distribution as $Z^{\boldsymbol{x}}$.

**ZMATMUL** If $x = W\bar{x}$ for $\bar{x} \in \boldsymbol{x}, W \in \mathcal{W}$ then $Z^{Wx} \stackrel{\text{def}}{=} \hat{Z}^{Wx} + \dot{Z}^{Wx}$, where:

    **ZHAT** $\hat{Z}^{Wx}$ is a Gaussian variable with zero mean. Let $\mathcal{V}_W$ denote the set of all vectors in the program of the form $Wy$ for some $y$. Then $\{\hat{Z}^{Wy} : Wy \in \mathcal{V}_W\}$ is defined to be jointly Gaussian with zero mean and covariance $\text{Cov}\left(\hat{Z}^{Wx}, \hat{Z}^{Wy}\right) \stackrel{\text{def}}{=} \sigma_W^2 \mathbb{E}\, Z^x Z^y$, for any $Wx, Wy \in \mathcal{V}_W$.. Furthermore, $\{\hat{Z}^{Wy} : Wy \in \mathcal{V}_W\}$ is mutually independent from $\{\hat{Z}^v : v \in \mathcal{V} \cup \bigcup_{\bar{W} \neq W} \mathcal{V}_{\bar{W}}\}$, where $\bar{W}$ ranges over $\mathcal{W} \cup \{A^\top : A \in \mathcal{W}\}$.

    **ZDOT** We can always unwind $Z^x = \Phi(\cdots)$, for some arguments $(\cdots) = (\{\hat{Z}^{W^\top y^i}\}_{i=1}^k, \{\hat{Z}^{z^i}\}_{i=1}^j; \{\mathring{\theta}_i\}_{i=1}^l)$, $z^i \notin \mathcal{V}_{W^\top}$ (where $\mathcal{V}_{W^\top}$ is defined in 5.3), and deterministic function $\Phi : \mathbb{R}^{k+j+l} \to \mathbb{R}$. Define $\partial Z^x / \partial \hat{Z}^{W^\top y^i} \stackrel{\text{def}}{=} \partial_i \Phi(\cdots)$. Then we set $\dot{Z}^{Wx} \stackrel{\text{def}}{=} \sigma_W^2 \sum_{i=1}^k Z^{y^i} \mathbb{E} \frac{\partial Z^x}{\partial \hat{Z}^{W^\top y^i}}$ There is some nuance in this definition, so see Remark A.1 and A.2.

The following theorem ties the symbolic nature of the $Z$s to the analytic nature of a Tensor Program.

**Theorem 5.4** (NE⊗ORT Master Theorem)**.** *Fix a* NE⊗ORT *program initialized accordingly to Setup 5.2. Assuming all nonlinearities are pseudo-Lipschitz in all arguments, then*

1. *For any collection of vectors $\boldsymbol{x} = \{x^1, ..., x^k\}$ and scalars $\Theta = \{\theta_1, ..., \theta_l\}$ in the program, and for any pseudo-Lipschitz $\psi : \mathbb{R}^{k(r+1)+l} \to \mathbb{R}$, as $n \to \infty$:*

$$\frac{1}{n^{r+1}} \sum_{\alpha, \beta_1, ..., \beta_r = 1}^n \psi(\boldsymbol{x}_\alpha, \boldsymbol{x}_{\beta_1}, ..., \boldsymbol{x}_{\beta_r}; \Theta) \xrightarrow{\text{a.s.}} \mathbb{E}_{Z_1^{\boldsymbol{x}}, Z_2^{\boldsymbol{x}}, ..., Z_{r+1}^{\boldsymbol{x}}}[\psi(Z_1^{\boldsymbol{x}}, Z_2^{\boldsymbol{x}}, ..., Z_{r+1}^{\boldsymbol{x}}; \mathring{\Theta})]$$

(10)

    *where $\{Z_j^{\boldsymbol{x}}\}_{j=1}^{r+1}$ are $r+1$ iid samples drawn from the same distribution as $Z^{\boldsymbol{x}}$, and $Z^{\boldsymbol{x}} = \{Z^{z^1}, ..., Z^{x^k}\}$ are defined according to Definition 5.3.*

2. *Any scalar $\theta$ in the program tends to $\mathring{\theta}$ almost surely such that $\forall_{p>0}, n^{1-p}(\theta - \mathring{\theta})^2 \xrightarrow{\text{a.s.}} 0$, where $\mathring{\theta}$ is as defined in Definition 5.3.*

Theorem 5.4 along with Definition 5.3 provide a general tool set to analyze adaptive (and standard) training of neural networks in the infinite-width limit, as long as the computational graph expressing the training process can be implemented in a NE⊗ORT program. Moreover, Theorem 5.4 provides a universal $O(n^{-1/2})$ asymptotic rate of convergence for all scalars produced by the program.

## 6 PROOF SKETCH

NE⊗ORT programs equipped with Theorem 5.4 provide the main tool to proving Theorem 4.1 and Theorem 4.2, and indeed their generalization to most common architectures and adaptive (and non adaptive) optimizers. This is done by adopting the following strategy: express the optimization dynamics using a NE⊗ORT program, mechanically compute the $Z$ variables according to Definition 5.3, and apply Theorem 5.4 to compute the limit (see proofs in appendix Appendix B). What remains is to prove Theorem 5.4 using a strategy which we now outline.

In a program, all vectors can be collected into an $n \times M$ matrix $V$ where $n$ is the dimension of each vector, and $M$ is the total number of vectors. The Master Theorem can be interpreted as saying that each row of $V$ (i.e., the slice for each $\alpha \in [n]$) is roughly an iid sample from some distribution $\mathcal{D}$ on $\mathbb{R}^M$ (which can be derived via the calculus of $Z$ random variables as in Definition 5.3). Specifically, Theorem 5.4 and all previous versions of the Master Theorem formalize this by saying: this matrix $V$ of vectors looks similar to a matrix $V'$ of iid samples from $\mathcal{D}$, as measured by applying arbitrary pseudo-Lipschitz "test function $\psi$" to both sides and taking averages.

**Core Insight:** Our core insight here is that $V$ is in fact similar to $V'$ in a stronger sense without needing to refer to any test function $\psi$: There is a "small" matrix $\delta V$ of the same size as $V$ such that $V - \delta V$ is distributed exactly the same as $V'$. In general, if this happens, then we say $V$ is *equivalent to $V'$*. The definition of "small" roughly means that each entry of $\delta V$ has typical size $O(n^{-1/2})$. Then, to recover Theorem 5.4, we just note that the test function $\psi$ is (by assumption) smooth enough that $\delta V$ contributes a vanishing amount to the LHS of Eq. (10).

To prove this core insight, there are two parts.

**Part 1:** We show that, in any NETSOR⊤ program (i.e., a program with no scalar variables and no TENSOR operation), $V$ is equivalent to $V'$. This can be done by re-analyzing the proof of the NETSOR⊤ Master Theorem in (Yang, 2020b) in a fairly straightforward way.

**Part 2:** For any NE⊗ORT program $\pi$ (the subject of our work here), we construct a *parallel NETSOR⊤ program* and show, by induction, that the vectors of the two programs are equivalent (i.e., distributed exactly the same after subtracting "small" vectors). This parallel program essentially replaces 1) all scalar variables in the original program by their deterministic limits, as computed in Definition 5.3(ZTENSORMOMENT), and 2) all TENSOR operations by NONLIN operations, as computed in Definition 5.3(ZTENSOR).

In this induction, we need to prove and use a lemma that, in the simplest case as an illustration, says the following: For any pseudo-Lipschitz function $\psi : \mathbb{R}^k \to \mathbb{R}$ and random vector $x \in \mathbb{R}^n$ with iid standard Gaussian entries, the following two tensors $T$ and $T'$ are equivalent: 1) the tensor $T$ with entries $T_{\beta_1...\beta_k} = \psi(x_{\beta_1}, \ldots, x_{\beta_k})$, and 2) the tensor $T'$ with entries $T'_{\beta_1...\beta_k} = \psi(x^1_{\beta_1}, \ldots, x^k_{\beta_k})$ where $x^1, \ldots, x^k$ are iid copies of $x$. The proof of this lemma interestingly requires Baranyai's theorem, a classical theorem from the theory of combinatorial design.

## 7 CONCLUSION

Adaptive optimizers are a staple in the modern deep learning toolkit, an are a necessary ingredient in most large scale neural network training. In this work, we have derived adaptive counterparts to the NTK and $\mu$ limit in prior works, which had only been derived for SGD. More generally, we have extended the Tensor Programs framework to allow the expression of any computation graph involving adaptive optimizers and the calculation of their large width limits. Our work lays a path to study the implicit bias of adaptive optimizers by studying their evolution equations in the infinite-width limit.

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

**Appendix organization**    The appendix is organized as follows:
In Appendix A we prove Theorem 5.4, which serves as the main tool to prove Theorem 4.1 and Theorem 4.2.
We then proceed to prove Theorem 4.1 and Theorem 4.2 in Appendix B. In Appendix C we extend the proofs of Theorem 4.1 and Theorem 4.2 to the case of AGOs with memory, and provide numerical verification to our results in Appendix D.

## A    FULL PROOF OF THEOREM 5.4

In this section we provide the proof for Theorem 5.4, restated:

**Theorem 5.4** (NE⊗ORT  Master Theorem). *Fix a* NE⊗ORT  *program initialized accordingly to Setup 5.2. Assuming all nonlinearities are pseudo-Lipschitz in all arguments, then*

1. *For any collection of vectors $\boldsymbol{x} = \{x^1, ..., x^k\}$ and scalars $\Theta = \{\theta_1, ..., \theta_l\}$ in the program, and for any pseudo-Lipschitz $\psi : \mathbb{R}^{k(r+1)+l} \to \mathbb{R}$, as $n \to \infty$:*

$$\frac{1}{n^{r+1}} \sum_{\alpha, \beta_1, ..., \beta_r = 1}^{n} \psi(\boldsymbol{x}_\alpha, \boldsymbol{x}_{\beta_1}, ..., \boldsymbol{x}_{\beta_r}; \Theta) \xrightarrow{\text{a.s.}} \mathbb{E}_{Z_1^{\boldsymbol{x}}, Z_2^{\boldsymbol{x}}, ..., Z_{r+1}^{\boldsymbol{x}}}[\psi(Z_1^{\boldsymbol{x}}, Z_2^{\boldsymbol{x}}, ..., Z_{r+1}^{\boldsymbol{x}}; \mathring{\Theta})]$$

   (10)

   *where $\{Z_j^{\boldsymbol{x}}\}_{j=1}^{r+1}$ are $r + 1$ iid samples drawn from the same distribution as $Z^{\boldsymbol{x}}$, and $Z^{\boldsymbol{x}} = \{Z^{z^1}, ..., Z^{x^k}\}$ are defined according to Definition 5.3.*

2. *Any scalar $\theta$ in the program tends to $\mathring{\theta}$ almost surely such that $\forall_{p>0}, n^{1-p}(\theta - \mathring{\theta})^2 \xrightarrow{\text{a.s.}} 0$, where $\mathring{\theta}$ is as defined in Definition 5.3.*

*Remark* A.1 (Partial derivative). The partial derivative in 5.3 should be interpreted as follows. By a simple inductive argument, $Z^x$ for every vector $x$ in the program is defined *uniquely* as a deterministic function $\varphi(\hat{Z}^{x^1}, \ldots, \hat{Z}^{x^k})$ of some $x^1, \ldots, x^k$ in $\mathcal{V}$ or introduced by MATMUL (notationally, we are suppressing the possible dependence on limit scalars $\mathring{\theta}_1, \ldots, \mathring{\theta}_l$). For instance, if in a program we have $A \in \mathcal{W}, v \in \mathcal{V}, y = Av, x = A^\top y$, then $Z^x = \hat{Z}^x + \hat{Z}^v$, so $\varphi$ is given by $\varphi(a, b) = a + b$. Then

$$\partial Z^x / \partial \hat{Z}^{x^i} \overset{\text{def}}{=} \partial_i \varphi(\hat{Z}^{x^1}, \ldots, \hat{Z}^{x^k}), \quad \text{and} \quad \partial Z^x / \partial \hat{Z}^z \overset{\text{def}}{=} 0 \text{ for any } z \notin \{x^1, \ldots, x^k\}.$$

Note this definition depends on the precise way the program is written, not just on the underlying mathematics. For example, if $y, z \in \mathcal{V}$ and $x = \phi(W(y + z))$, then $Z^x = \phi(\hat{Z}^{W(y+z)})$ so that $\partial Z^x / \partial \hat{Z}^{Wy} = \partial Z^x / \partial \hat{Z}^{Wz} = 0$. If instead, we have $x = \phi(Wy + Wz)$, then $Z^x = \phi(\hat{Z}^{Wy} + \hat{Z}^{Wz})$ so that $\partial Z^x / \partial \hat{Z}^{W(x+y)} = 0$. However, in both cases, $\dot{Z}^{W^\top x} = (Z^y + Z^z) \mathbb{E}\, \phi'(\hat{Z}^{W(y+z)})$.

*Remark* A.2 (Partial derivative expectation). The quantity $\mathbb{E}\, \frac{\partial Z^x}{\partial \hat{Z}^{W^\top y}}$ is well defined if $Z^x$ is differentiable in $\hat{Z}^{W^\top y}$. However, even if this is not the case, e.g. if $x = \theta(W^\top y)$ where $\theta$ is the Heavyside step function, we can still define this expectation by leveraging Stein's lemma:

In 5.3, suppose $\{W^\top y^i\}_{i=1}^k$ are all elements of $\mathcal{V}_{W^\top}$ introduced before $x$. Define the matrix $C \in \mathbb{R}^{k \times k}$ by $C_{ij} \overset{\text{def}}{=} \mathbb{E}\, Z^{y^i} Z^{y^j}$ and define the vector $b \in \mathbb{R}^k$ by $b_i \overset{\text{def}}{=} \mathbb{E}\, \hat{Z}^{W^\top y^i} Z^x$. If $a = C^+ b$ (where $C^+$ denotes the pseudoinverse of $C$), then in 5.3 we may set

$$\sigma_W^2 \mathbb{E}\, \frac{\partial Z^x}{\partial \hat{Z}^{W^\top y^i}} = a_i.$$

   (11)

This definition agrees with the partial derivative expectation by Stein's lemma when the latter is well defined.

**Pseudo-Lipschitz functions**    are, roughly speaking, functions whose weak derivatives are polynomially bounded.

**Definition A.3.** A function $f : \mathbb{R}^k \to \mathbb{R}$ is called *pseudo-Lipschitz* of degree $d$ if $|f(x) - f(y)| \le C\|x - y\|(1 + \sum_{i=1}^k |x_i|^d + |y_i|^d)$ for some $C$. We say $f$ is pseudo-Lipschitz if it is so for any degree.

Here are some basic properties of pseudo-Lipschitz functions:

- The norm $\| \cdot \|$ in Definition A.3 can be any norm equivalent to the $\ell_2$ norm, e.g. $\ell_p, p \geq 1$, norms. Similarly, $\sum_{i=1}^{k} |x_i|^d + |y_i|^d$ can be replaced by $\|x\|_p^d + \|y\|_p^d$, for any $p \geq 1$.

- A pseudo-Lipschitz function is polynomially bounded.

- A composition of pseudo-Lipschitz functions of degrees $d_1$ and $d_2$ is pseudo-Lipschitz of degree $d_1 + d_2$.

- A pseudo-Lipschitz function is Lipschitz on any compact set.

**Indexing notations**    We use superscripts to distinguish between different tensors in the program, and subscripts to index into coordinates of tensors. (i.e $x_j^i$ denotes the $j$'th coordinate of vector $x^j \in \mathbb{R}^n$. We typically use $\boldsymbol{\beta} = \{\beta_1, \beta_2, ..., \beta_r\}$ to denote a set of coordinates (typically containing $r$ coordinates unless specified otherwise). For any vector $x \in \mathbb{R}^n$, $x_{\boldsymbol{\beta}}$ denotes the set $\{x_{\beta_1}, x_{\beta_2}, ..., x_{\beta_r}\}$. For summation over all indices in $\boldsymbol{\beta}$, we use the abbreviated notation $\sum_{\boldsymbol{\beta}=1}^{n} \stackrel{\text{def}}{=} \sum_{\beta_1=1}^{n} \cdots \sum_{\beta_r=1}^{n}$. We typically use $\boldsymbol{x}$ to denote a set of vectors $\{x^1, ..., x^k\}$, where the size of the set is implied by the context. The notation $\boldsymbol{x}_{\boldsymbol{\beta}}$ refers to the set of scalars $\{x_{\beta_1}^1, ..., x_{\beta_1}^k, ..., x_{\beta_r}^1, ..., x_{\beta_r}^k\}$ (note that in this case $|\boldsymbol{x}_{\boldsymbol{\beta}}| = kr$. We additionally use $\alpha$ to index into tensors. The notation $x_{\alpha,\boldsymbol{\beta}}$ for a vector $x$ refers to the set $\{x_\alpha, x_{\beta_1}, ..., x_{\beta_r}\}$. Similarly, the notation $\boldsymbol{x}_{\alpha,\boldsymbol{\beta}}$ for $\boldsymbol{x} = \{x^1, ..., x^k\}$ refers to the set of scalars $\{x_\alpha^1, ..., x_\alpha^k, x_{\beta_1}^1, ..., x_{\beta_1}^k, ..., x_{\beta_r}^1, ..., x_{\beta_r}^k\}$ (in this case $|\boldsymbol{x}_{\alpha,\boldsymbol{\beta}}| = k(r+1)$). We use $c, C, \tilde{C}$ as arbitrary constant scalars throughout the appendix (Their value might change between in different lines).

*Proof.* As in previous versions of *Tensor Programs*, we prove Theorem 5.4 by inducting on the vectors and scalars in the program. We use the following definitions throughout the proof:

**Definition A.4.** For any set of vectors $\boldsymbol{x} = \{x^1, x^2, ..., x^k\}$ in the program, define the matrix $\Lambda^{\boldsymbol{x}} \in \mathbb{R}^{k \times k} : \Lambda_{\alpha,\beta}^{\boldsymbol{x}} = \frac{x^{\alpha\top} x^\beta}{n}$. We say the set $\boldsymbol{x}$ has stable rank if $\lim_{n \to \infty} \text{rank}(\Lambda^{\boldsymbol{x}}) \stackrel{\text{a.s}}{=} \text{rank}(\lim_{n \to \infty} \Lambda^{\boldsymbol{x}})$.

**Definition A.5.** We say a random vector $x \in \mathbb{R}^n$ is vanishing if $\forall_{p>0} \lim_{n \to \infty} \frac{\|x\|^2}{n^p} \stackrel{\text{a.s}}{=} 0$.

**Definition A.6.** We say a random vector $x \in \mathbb{R}^n$ is regular with constants $\{\check{c}(p), \hat{c}(p)\}$ if for all $p \in (0, \infty)$ there exists constants $0 < \check{c}(p) \leq \hat{c}(p) < \infty$ such that $\check{c}(p) \leq \lim_{n \to \infty} \frac{\|x\|_p^p}{n} \leq \hat{c}(p)$ almost surely.

**Definition A.7.** We say a random scalar $\theta \in \mathbb{R}$ is vanishing if it converges almost surely to $\mathring{\theta}$, and it holds that $\forall_{p>0}, \ \lim_{n \to \infty} \frac{(\theta - \mathring{\theta})^2}{n^{p-1}} \stackrel{\text{a.s}}{=} 0$. Equivalently, $(\theta - \mathring{\theta})\mathbf{1}_n$ is a vanishing vector.

Definition A.5 and Definition A.6 extend naturally to tensors, which will come in handy in our analysis. We index a rank $r$ tensor $x \in \otimes^r \mathbb{R}^n$ with indices $\boldsymbol{\beta} = \{\beta_1, ..., \beta_r\}$, such that $x_{\boldsymbol{\beta}} = x[\beta_1, \beta_2, ..., \beta_r]$.

**Definition A.8.** We say a rank $r$ random tensor $x \in \otimes^r \mathbb{R}^n$ is vanishing if $\forall_{p>0} \lim_{n \to \infty} \frac{\|x\|^2}{n^p} \stackrel{\text{a.s}}{=} 0$.

**Definition A.9.** We say a rank $r$ random tensor $x \in \otimes^r \mathbb{R}^n$ is regular with constants $\{\check{c}(p), \hat{c}(p)\}$ if for all $p \in (0, \infty)$ there exists constants $0 < \check{c}(p) \leq \hat{c}(p) < \infty$ such that $\check{c}(p) \leq \lim_{n \to \infty} \frac{\|x\|_p^p}{n^r} \leq \hat{c}(p)$ almost surely.

## A.1    INDUCTION HYPOTHESIS

**Setup A.10** (Induction Setup). *We will keep track of a set of vanishing scalars $\Theta$ (see Definition A.7), and two sets of vectors: The core set $\boldsymbol{x}$ which contains regular vectors produced by a* MATMUL *operation (see Definition A.6 and Definition 5.1), and a vanishing set $\hat{\boldsymbol{x}}$ of vanishing vectors (see Definition A.5). We will denote by $\boldsymbol{x}_W$ the set of vectors $y : Wy \in \boldsymbol{x}$. Given the sets of vanishing*

scalars $\Theta$, corset $\boldsymbol{x}$ and vanishing set $\hat{\boldsymbol{x}}$ at some step $m$ in the program, let $h \in \mathbb{R}^n, \theta \in \mathbb{R}$ define a new vector and scalar via TENSOR and TENSORMOMENT operations respectively. Namely:

$$h_\alpha = \frac{1}{n^r} \sum_{\boldsymbol{\beta}=1}^n \psi(\boldsymbol{x}_{\alpha,\boldsymbol{\beta}}; \hat{\boldsymbol{x}}_{\alpha,\boldsymbol{\beta}}; \Theta), \quad \theta = \frac{1}{n} \sum_{\alpha=1}^n h_\alpha \tag{12}$$

where $\psi : \mathbb{R}^{(|\boldsymbol{x}|+|\hat{\boldsymbol{x}}|)(r+1)+l} \to \mathbb{R}$ is pseudo lipschitz in all of its arguments. We further define:

$$h_\alpha^0 = \frac{1}{n^r} \sum_{\boldsymbol{\beta}=1}^n \psi(\boldsymbol{x}_{\alpha,\boldsymbol{\beta}}; \mathbf{0}; \mathring{\Theta}) \tag{13}$$

$$\bar{h}_\alpha = \mathbb{E}_{Z_1^{\boldsymbol{x}}, Z_2^{\boldsymbol{x}}, ..., Z_r^{\boldsymbol{x}}} \big[ \psi(\boldsymbol{x}_\alpha, Z_1^{\boldsymbol{x}}, Z_2^{\boldsymbol{x}}, ..., Z_r^{\boldsymbol{x}}; \mathbf{0}; \mathring{\Theta}) \big] \tag{14}$$

$$\Delta h = h - x^0 \tag{15}$$

$$\Delta \bar{h} = h^0 - \bar{h} \tag{16}$$

Our induction hypothesis $\mathbf{IH}(m)$ asserts that the following hold simultaneously:

1. **ReWrite**$(m)$ Any vector produced by MATMUL can be written as a linear combination of vectors from $\boldsymbol{x}$ and $\bar{\boldsymbol{x}}$.

2. **StableRank**$(m)$ For any $W \in \mathbb{R}^{n \times n}$, the set $\boldsymbol{x}_W$ has a stable rank.

3. **Dichotomy**$(m)$ It holds that:

   (a) $\bar{h}$ is either a regular or a vanishing vector.
   (b) $\Delta h, \Delta \bar{h}$ are vanishing vectors.

4. **TensorMoment**$(m)$ It holds that $\theta \overset{\text{a.s}}{\to} \mathring{\theta}$, where:

$$\mathring{\theta} = \mathbb{E}_{Z_1^{\boldsymbol{x}}, ..., Z_{r+1}^{\boldsymbol{x}}} \big[ \psi(Z_1^{\boldsymbol{x}}, ..., Z_{r+1}^{\boldsymbol{x}}; \mathbf{0}; \mathring{\Theta}) \big] \tag{17}$$

5. **ConvRate**$(m)$ It holds that $\theta \in \mathbb{R}$ is a vanishing scalar, or:

$$\forall_{p>0}, \quad \frac{(\theta - \mathring{\theta})^2}{n^{p-1}} \overset{\text{a.s}}{\to} 0 \tag{18}$$

## A.2 HELPER LEMMAS

We use the following Lemmas regularly throughout the proof. Note that some of the lemmas are stated for the vector, however their extension to tensors are immediate.

**Lemma A.11.** *If $u, v \in \mathbb{R}^n$ are vanishing vectors, then $\nu = u + v$ is a vanishing vector.*

*Proof.*

$$\forall_{p>0}, \quad \frac{\|\nu\|^2}{n^p} = \frac{\|u\|^2 + \|v\|^2 + 2u \odot v}{n^p} \leq 3 \frac{\|u\|^2}{n^p} + 3 \frac{\|v\|^2}{n^p} \overset{\text{a.s}}{\to} 0 \tag{19}$$

$\square$

**Lemma A.12.** *If $u, v \in \mathbb{R}^n$ are regular vectors, then $\nu = |u| + |v|$ is a regular vector.*

*Proof.* for $p \in [1, \infty)$, using triangle inequality for $p$ norms:

$$\forall_{p \geq 1}, \quad \frac{\|\nu\|_p^p}{n} = \frac{\||u| + |v|\|_p^p}{n} \leq \left( \big( \frac{\|u\|_p^p}{n} \big)^{\frac{1}{p}} + \big( \frac{\|v\|_p^p}{n} \big)^{\frac{1}{p}} \right)^p \overset{\text{a.s}}{\leq} \left( \hat{c}_u(p)^{\frac{1}{p}} + \hat{c}_v(p)^{\frac{1}{p}} \right)^p \tag{20}$$

For $p \in (0, 1)$:

$$\forall_{0<p<1}, \quad \frac{\|\nu\|_p^p}{n} = \frac{\||u| + |v|\|_p^p}{n} \leq \frac{\|u\|_p^p}{n} + \frac{\|v\|_p^p}{n} \overset{\text{a.s}}{\leq} \hat{c}_u(p) + \hat{c}_v(p) \tag{21}$$

One the other hand the lower bound is trivially $\forall_{p \geq 1}, \frac{\|\nu\|_p^p}{n} \overset{\text{a.s}}{\geq} \max(\check{c}_u(p), \check{c}_v(p))$.

$\square$

**Lemma A.13.** *If $u \in \mathbb{R}^n$ is a vanishing vector, then for any $p > 0$, it holds that $\frac{1}{n}\|u\|_p^p \overset{a.s}{\to} 0$ (i.e $u$ has vanishing moments).*

*Proof.* For $p \geq 2$, we have that:

$$\frac{\|u\|_p^p}{n} \leq \frac{\|u\|^p}{n} = \left(\frac{\|u\|^2}{n^{\frac{2}{p}}}\right)^{\frac{p}{2}} \overset{a.s}{\to} 0 \tag{22}$$

For $0 < p < 2$, using the fact that $\forall_{0<p<q}, \ \|u\|_p \leq n^{\frac{1}{p}-\frac{1}{q}}\|u\|_q$ and assigning $q = 2$:

$$\frac{\|u\|_p^p}{n} \leq \frac{n^{1-\frac{p}{2}}\|u\|^p}{n} = \left(\frac{\|u\|^2}{n}\right)^{\frac{p}{2}} \overset{a.s}{\to} 0 \tag{23}$$

which proves the claim. $\qquad\square$

**Lemma A.14.** *If $u \in \mathbb{R}^n$ is a vanishing vector, and $v \in \mathbb{R}^n$ is a regular vector, then $\nu = u + v$ is a regular vector.*

*Proof.* This is immediate from Lemma A.13 and Lemma A.12, and setting the constants for $u$ $\{\check{c}_u(p), \hat{c}_u(p)\} = \{0, 0\}$. $\qquad\square$

**Lemma A.15.** *If $u \in \mathbb{R}^n$ is a vanishing vector, then for any $r \geq 1$, $\nu = |u|^r$ is a vanishing vector. If $u \in \mathbb{R}^n$ is a regular vector, then for any $r \geq 0$, $\nu = |u|^r$ is a regular vector.*

*Proof.* For vanishing $u$, using elementary norm bounds, that $\forall_{1<p<q}, \ \|u\|_q \leq \|u\|_p$ and assigning $q = 2$:

$$\forall_{p>0}, \ \frac{\|\nu\|^2}{n^p} = \frac{\|u\|_{2r}^{2r}}{n^p} \leq \frac{\|u\|^{2r}}{n^p} = \left(\frac{\|u\|^2}{n^{\frac{p}{r}}}\right)^r \overset{a.s}{\to} 0 \tag{24}$$

The proof for regular $u$ follows immediately from the definition of a regular vector. $\qquad\square$

**Lemma A.16.** *If $u \in \mathbb{R}^n$ is a vanishing vector, and $v \in \mathbb{R}^n$ is a regular vector with constants $\{\check{c}(p), \hat{c}(p)\}$, then $\nu = u \odot v$ is a vanishing vector.*

*Proof.* For any $p > 0$, choose $m, l \in (0, 1)$ such that $p > m$, and $l + m = 1$. Using Holders inequality:

$$\forall_{p\geq 0}, \ \frac{\|\nu\|^2}{n^p} = \sum_{i=1}^{n} \frac{u_i^2}{n^{p-m}} \frac{v_i^2}{n^m} \leq \left(\sum_{i=1}^{n} \frac{|u_i|^{\frac{2}{l}}}{n^{\frac{p-m}{l}}}\right)^l \left(\sum_{i=1}^{n} \frac{|v_i|^{\frac{2}{m}}}{n}\right)^m \tag{25}$$

$$\leq \left(\frac{\||u|^{\frac{1}{l}}\|^2}{n^{\frac{p-m}{l}}}\right)^l \left(\sum_{i=1}^{n} \frac{|v_i|^{\frac{2}{m}}}{n}\right)^m = \left(\frac{\||u|^{\frac{1}{l}}\|^2}{n^{\frac{p-m}{l}}}\right)^l \left(\frac{\|v\|_{\frac{2}{m}}^{\frac{2}{m}}}{n}\right)^m \overset{a.s}{\leq} 0 \cdot \left(\hat{c}(\frac{2}{m})\right)^m = 0 \tag{26}$$

where we used Lemma A.15 to assert that $\frac{\||u|^{\frac{1}{l}}\|^2}{n^{\frac{p-m}{l}}} \overset{a.s}{\to} 0$. $\qquad\square$

**Lemma A.17.** *If $\theta$ is a vanishing scalar, then $f(\theta)$ for $f : \mathbb{R} \to \mathbb{R}$ is a vanishing scalar if $f$ is locally lipschitz at $\mathring{\theta}$, and $f(\mathring{\theta}) = 0$.*

*Proof.* WLOG assume $\forall_{(\theta-\mathring{\theta})^2 < \epsilon} f(\theta) < A|\theta - \mathring{\theta}|$ for some $\epsilon \in \mathbb{R}$. Define :

$$g(\theta) = \begin{cases} A|\theta - \mathring{\theta}| & (\theta - \mathring{\theta})^2 \leq \epsilon \\ f(\theta) & \text{else} \end{cases} \tag{27}$$

Since $\theta \to \mathring{\theta}$ almost surely, this implies that $\text{Prob}(\lim_{n\to\infty}(\theta - \mathring{\theta})^2 < \epsilon) = 1$, hence $(\theta - \mathring{\theta})^2 < \epsilon$ almost surely. Therefore:

$$\forall_{p>0}, \ \lim_{n\to\infty} \frac{f(\theta)}{n^{1-p}} \leq \lim_{n\to\infty} \frac{g(\theta)}{n^{1-p}} \overset{a.s}{=} \lim_{n\to\infty} A\frac{(\theta - \mathring{\theta})^2}{n^{1-p}} \overset{a.s}{=} 0 \tag{28}$$

$\qquad\square$

**Lemma A.18.** *Let $x, \hat{x}$ denote sets of regular and vanishing vectors, and let $\psi : \mathbb{R}^{|x|+|\hat{x}|} \to \mathbb{R}$ be pseudo lipschitz. Then:*

$$\lim_{n \to \infty} \frac{1}{n} \sum_{\alpha=1}^{n} \psi(x_\alpha; \hat{x}_\alpha) = \lim_{n \to \infty} \frac{1}{n} \sum_{\alpha=1}^{n} \psi(x_\alpha; 0) \tag{29}$$

*Proof.* We have that:

$$\psi(x_\alpha; 0) - \left| \psi(x_\alpha; \hat{x}_\alpha) - \psi(x_\alpha; 0) \right| \le \psi(x_\alpha; \hat{x}_\alpha) \le \psi(x_\alpha; 0) + \left| \psi(x_\alpha; \hat{x}_\alpha) - \psi(x_\alpha; 0) \right| \tag{30}$$

Since $\psi$ is pseudo lipschitz:

$$\left| \psi(x_\alpha; \hat{x}_\alpha) - \psi(x_\alpha; 0) \right| \le \|\hat{x}_\alpha\| \left( 1 + \|\hat{x}_\alpha\|_d^d + \|x_\alpha\|_d^d \right) \tag{31}$$

Define the vectors $u_\alpha = \left( 1 + \|\hat{x}_\alpha\|_d^d + \|x_\alpha\|_d^d \right), v_\alpha = \|\hat{x}_\alpha\|$ Note that $u, v$ are regular and venishing vectors respectively, hence by Lemma A.16 $u \odot v$ is a vanishing vector. Then by Lemma A.13, $\frac{1}{n} \sum_{\alpha=1}^{n} (u \odot v)_\alpha \to 0$ almost surely. Plugging into Eq. (30) and taking the limit, we have that:

$$\lim_{n \to \infty} \frac{1}{n} \sum_{\alpha=1}^{n} \psi(x_\alpha; 0) - 0 \le \lim_{n \to \infty} \frac{1}{n} \sum_{\alpha=1}^{n} \psi(x_\alpha; \hat{x}_\alpha) \le \lim_{n \to \infty} \frac{1}{n} \sum_{\alpha=1}^{n} \psi(x_\alpha; 0) + 0 \tag{32}$$

proving the claim. $\square$

Note that Lemmas A.11 to A.16 and A.18 trivially extend to vanishing and regular tensors.

**Lemma A.19.** *If $u \in \otimes^{r+1} \mathbb{R}^n, r \ge 1$ is a vanishing tensor, then $\nu \in \mathbb{R}^n : \nu_\alpha = \frac{1}{n^{\frac{r}{2}}} \sum_{\beta=1}^{n} u_{\alpha, \beta}$ is a vanishing vector.*

*Proof.* Using the elementary inequality $\|v\|_1 \le \sqrt{n}\|v\|$ for any vector $v \in \mathbb{R}^n$, we have:

$$\forall_{p>0}, \frac{\|\nu\|^2}{n^p} \le \frac{\sum_{\alpha=1}^{n} \left( \sum_{\beta=1}^{n} |u_{\alpha, \beta}| \right)^2}{n^{r+p}} \le \frac{\sum_{\alpha=1}^{n} \left( \sqrt{n^r} \sqrt{\sum_{\beta=1}^{n} u_{\alpha, \beta}^2} \right)^2}{n^{r+p}} \tag{33}$$

$$= \frac{n^r \sum_{\alpha, \beta=1}^{n} u_{\alpha, \beta}^2}{n^{r+p}} = \frac{\sum_{\alpha, \beta=1}^{n} u_{\alpha, \beta}^2}{n^p} \xrightarrow{\text{a.s}} 0 \tag{34}$$

$\square$

**Lemma A.20.** *Let $\{x_n\}_{n>0}$ be a sequence of random variables. If for some $t \in \mathbb{N}$, and for all $n$ it holds that $\mathbb{E}x_n^{2t} \le cn^{-1-\lambda}$ for $c, \lambda > 0$ then $x_n \to 0$ almost surely.*

*Proof.* by markov's inequality, for any $\epsilon > 0$:

$$P(|x_n| > \epsilon) = P(x_n^{2t} > \epsilon^{2t}) \le \frac{\mathbb{E}x_n^{2t}}{\epsilon^{2t}} \tag{35}$$

$$\sum_{n=1}^{\infty} P(|x_n| > \epsilon) \le \sum_{n=1}^{\infty} \frac{c}{\epsilon^{2t} n^{1+\lambda}} < \infty \tag{36}$$

By the Borel-Cantelli Lemma, $x_n \to 0$ almost surely. $\square$

**Lemma A.21.** *Let $m, n, r \in \mathbb{N}$ such that $n$ is divisible by $r$. Let $\{\nu^i\}_{i=1}^{m}, \forall_i, \nu^i \in \mathbb{R}^{\frac{n}{r}}$ denote random (possibly dependent), zero mean vectors with iid coordinates and finite moments of any order $\forall_{q \in \mathbb{N}}, \mathbb{E}[(\nu_\alpha^i)^{2q}] = C_{2q}$. Define $S^i = \frac{1}{\sqrt{n}} \sum_{\alpha=1}^{\frac{n}{r}} \nu_\alpha^i$. Then, there exists a function $f(q) \in [0, \infty)$ independent on $n, m$ such that:*

$$\mathbb{E}\left[ \left( \sum_{i=1}^{m} \frac{S^i}{m} \right)^{2q} \right] \le f(q) C_{2q} \tag{37}$$

*Proof.* Using Hölder's inequality:

$$\mathbb{E}\Big[\sum_{i=1}^{m}\frac{S^i}{m}\Big]^{2q} = \frac{1}{m^{2q}}\sum_{\beta_1,...,\beta_{2q}=1}^{m}\mathbb{E}\Big[\prod_{l=1}^{2q}S^{\beta_l}\Big] \tag{38}$$

$$\leq \frac{1}{m^{2q}}\sum_{\beta_1,...,\beta_{2q}=1}^{m}\sqrt[2q]{\prod_{l=1}^{2q}\mathbb{E}\big[(S^{\beta_l})^{2q}\big]} \tag{39}$$

The inner expectations are given by:

$$\mathbb{E}\big[(S^i)^{2q}\big] = \mathbb{E}\Big[\Big(\sum_{\alpha=1}^{\frac{n}{r}}\frac{\nu_\alpha^i}{\sqrt{n}}\Big)^{2q}\Big] = \frac{1}{n^q}\sum_{\alpha_1,...,\alpha_{2q}=1}^{\frac{n}{r}}\mathbb{E}\Big[\prod_{j=1}^{2q}\nu_{\alpha_j}^i\Big] \tag{40}$$

Note that since the random variables $\nu_\alpha^i$ have zero mean, the expectation $\mathbb{E}\Big[\prod_{j=1}^{2q}\nu_{\alpha_j}^i\Big]$ does not vanish only when the indices $\alpha_1,...,\alpha_{2q}$ do not contain an entry which appears in isolation. In other words, the number of non-zero terms $n^\star$ in the sum is:

$$n^\star = \sum_{\substack{\{u_i\}_{i=1}^{q}\in\mathbb{N} \\ u_q\geq u_{q-1}\geq...\geq u_1 \\ \forall_i, u_i\neq 1 \\ \sum_{i=1}^{q}u_i=2q}} \binom{2q}{u_q}\binom{2q-u_q}{u_{q-1}}...\binom{2q-\sum_{i=2}^{q}u_i}{u_1}\frac{\frac{n}{r}!}{(\frac{n}{r}-\sum_{i=1}^{q}\mathbb{1}_{u_i>0})!} \tag{41}$$

Note that the the only term which depends on $n$ is $\frac{\frac{n}{r}!}{(\frac{n}{r}-\sum_{i=1}^{q}\mathbb{1}_{u_i>0})!}$, which is bounded by:

$$\frac{\frac{n}{r}!}{(\frac{n}{r}-\sum_{i=1}^{q}\mathbb{1}_{u_i>0})!} \leq \frac{\frac{n}{r}!}{(\frac{n}{r}-q)!} \leq (\frac{n}{r})^q\tilde{f}(q) \tag{42}$$

where $\tilde{f}(q) < \infty$ independent on $n, m$. It follows:

$$n^\star \leq (\frac{n}{r})^q\tilde{f}(q) \sum_{\substack{\{u_i\}_{i=1}^{q}\in\mathbb{N} \\ u_q\geq u_{q-1}\geq...\geq u_1 \\ \forall_i, u_i\neq 1 \\ \sum_{i=1}^{q}u_i=2q}} \binom{2q}{u_q}\binom{2q-u_q}{u_{q-1}}...\binom{2q-\sum_{i=2}^{q}u_i}{u_1} \tag{43}$$

$$\leq \frac{n^q}{r}f(q) \quad \text{where:} \tag{44}$$

$$f(q) \stackrel{\text{def}}{=} \tilde{f}(q) \sum_{\substack{\{u_i\}_{i=1}^{q}\in\mathbb{N} \\ u_q\geq u_{q-1}\geq...\geq u_1 \\ \forall_i, u_i\neq 1 \\ \sum_{i=1}^{q}u_i=2q}} \binom{2q}{u_q}\binom{2q-u_q}{u_{q-1}}...\binom{2q-\sum_{i=2}^{q}u_i}{u_1} \tag{45}$$

In addition, from Hölder's inequality:

$$\mathbb{E}\Big[\prod_{j=1}^{2q}\nu_{\alpha_j}^i\Big] \leq \sqrt[2q]{\prod_{j=1}^{2q}\mathbb{E}\big[(\nu_{\alpha_j}^i)^{2k}\big]} = C_{2q} \tag{46}$$

Hence it follows that $\mathbb{E}\big[(S^i)^{2q}\big] \leq \frac{(\frac{n}{r})^q f(q)}{n^q}C_{2q} = \frac{f(q)}{r^q}C_{2q}$. Inserting into Eq. (38), we finally get:

$$\mathbb{E}\big[(\sum_{i=1}^{m}\frac{S^i}{m})^{2q}\big] \leq \frac{1}{m^{2q}}\sum_{\beta_1,...,\beta_{2q}=1}^{m}\sqrt[2q]{\prod_{l=1}^{2q}\big[\frac{f(q)}{r^q}C_{2q}\big]} \tag{47}$$

$$\leq \frac{f(q)}{r^q}C_{2q} \leq f(q)C_{2q} \tag{48}$$

$\square$

Before delving into the full proof of the induction hypothesis, we note the following fact that immediately holds at any arbitrary step in the induction. Let $h, h^0, \Delta h, \theta$ be defined as in Setup A.10.

Then the following claim holds:

**Claim A.1.** $\Delta h$ *is a vanishing vector.*

*Proof.* Since $\psi$ is psuedo lipshitz, there exists some $d \geq 0$ such that:

$$\left|\Delta h_\alpha\right| = \frac{1}{n^r}\left|\sum_{\boldsymbol{\beta}=1}^{n}\left(\psi(\boldsymbol{x}_{\alpha,\boldsymbol{\beta}}; \hat{\boldsymbol{x}}_{\alpha,\boldsymbol{\beta}}; \Theta) - \psi(\boldsymbol{x}_{\alpha,\boldsymbol{\beta}}; \mathbf{0}; \mathring{\Theta}))\right| \tag{49}$$

$$\leq \frac{1}{n^r}\sum_{\boldsymbol{\beta}=1}^{n}\sqrt{\|\Theta - \mathring{\Theta}\|^2 + \|\hat{\boldsymbol{x}}_\alpha\|^2 + \|\hat{\boldsymbol{x}}_{\boldsymbol{\beta}}\|^2}\Big(1 + \|\Theta\|_d^d + \|\mathring{\Theta}\|_d^d + \|\boldsymbol{x}_{\boldsymbol{\beta}}\|_d^d + \|\hat{\boldsymbol{x}}_{\boldsymbol{\beta}}\|_d^d\dots \tag{50}$$

$$+ \|\boldsymbol{x}_\alpha\|_d^d + \|\hat{\boldsymbol{x}}_\alpha\|_d^d\Big) \tag{51}$$

$$\leq \frac{1}{n^{\frac{r}{2}}}\sum_{\boldsymbol{\beta}=1}^{n}\tau_{\alpha,\boldsymbol{\beta}}T_{\alpha,\boldsymbol{\beta}} \tag{52}$$

where we have defined the tensors $\tau, T \in \otimes^{r+1}\mathbb{R}^n$ such that:

$$T_{\alpha,\boldsymbol{\beta}} = 1 + \|\Theta\|_d^d + \|\mathring{\Theta}\|_d^d + \|\boldsymbol{x}_{\boldsymbol{\beta}}\|_d^d + \|\hat{\boldsymbol{x}}_{\boldsymbol{\beta}}\|_d^d + \|\boldsymbol{x}_\alpha\|_d^d + \|\hat{\boldsymbol{x}}_\alpha\|_d^d \tag{53}$$

$$\tau_{\alpha,\boldsymbol{\beta}} = \frac{\|\Theta - \mathring{\Theta}\|_1 + \|\hat{\boldsymbol{x}}_{\boldsymbol{\beta}}\|_1 + \|\hat{\boldsymbol{x}}_\alpha\|_1}{n^{\frac{r}{2}}} \tag{54}$$

Note that by Lemmas A.11 to A.15, $T$ is a regular tensor, while $\tau$ is a vanishing tensor. Hence, $T \odot \tau$ is a vanishing tensor by Lemma A.16, hence $\Delta h$ is a vanishing vector by Lemma A.19. $\qquad\square$

Claim A.1 is useful due to the following general claim:

**Claim A.2.** *If **Dichotomy**$(m)$,**TensorMoment**$(m)$ and **ConvRate**$(m)$ apply for the vector $h$ (as defined in Setup A.10), then it applies for $h + \delta$ if $\delta$ is a vanishing vector.*

*Proof.* This is true due to the function $\psi$ being pseudo lipschitzness. More specifically:

1. If **Dichotomy**$(m)$ holds for $h$, then it holds for $h + \delta$ This is trivially true due since we can expand $\hat{\boldsymbol{x}} := \hat{\boldsymbol{x}} \cup \delta$, and invoke Lemmas A.11 and A.14.

2. If **TensorMoment**$(m)$ holds for $h$, then it holds for $h + \delta$. This is since we may trivially assert that: $\theta = \frac{1}{n}\sum_{\alpha=1}^{n}(h_\alpha + \delta_\alpha) \to \frac{1}{n}\sum_{\alpha=1}^{n}h_\alpha$, which stems from Lemma A.13.

3. If **ConvRate**$(m)$ holds for $h$, then it holds for $h + \delta$. This is since:

$$\forall_{p>\epsilon>0}, \frac{1}{n^{p-1}}(\frac{1}{n}\sum_{\alpha=1}^{n}\delta_\alpha + \theta - \mathring{\theta})^2 = \frac{1}{n^{p-1}}(\frac{1}{n}\sum_{\alpha=1}^{n}\delta_\alpha)^2 + \frac{2}{n^p}\sum_{\alpha=1}^{n}\delta_\alpha(\theta - \mathring{\theta}) \tag{55}$$

$$+ \frac{1}{n^{p-1}}(\theta - \mathring{\theta})^2 \tag{56}$$

$$\leq (\frac{\|\delta\|_1}{n^{\frac{p+1}{2}}})^2 + 2\sqrt{\frac{n(\theta - \mathring{\theta})^2}{n^{\frac{p-\epsilon}{2}}}}\sqrt{\frac{\|\delta\|^2}{n^{\frac{\epsilon}{2}}}} + \frac{1}{n^{p-1}}(\theta - \mathring{\theta})^2 \tag{57}$$

$$\leq \frac{\|\delta\|^2}{n^p} + 2\sqrt{\frac{(\theta - \mathring{\theta})^2}{n^{\frac{p-\epsilon}{2}-1}}}\sqrt{\frac{\|\delta\|^2}{n^{\frac{\epsilon}{2}}}} + \frac{(\theta - \mathring{\theta})^2}{n^{p-1}} \overset{\text{a.s}}{\to} 0 \tag{58}$$

$\qquad\square$

Given Claim A.2 and Eq. (53), we may prove **Dichotomy**$(m)$,**TensorMoment**$(m)$ and **ConvRate**$(m)$ for $h^0$ instead of $h$.

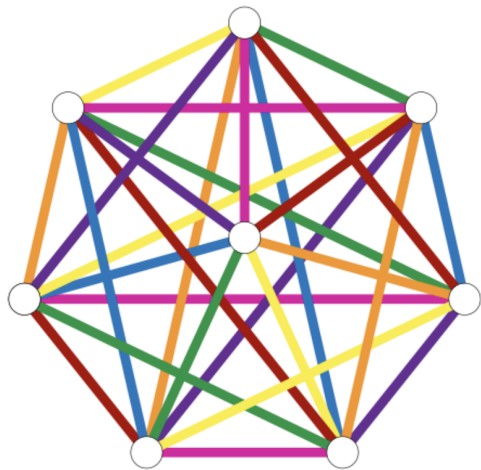

Figure 1: **Baranyai's Theorem.** A graphical illustration of *Baranyai's Theorem for $n = 8, r = 2$.* A partition of 8 vertices into 1 factors, represented by different colors. Each 1 factor is a partition of the vertices into hyperedges (in this case, since $r = 2$, simply edges) where no vertex is shared between two edges, and no edge is shared between two 1 factors. Baranyai's Theorem states that there are $\binom{8}{2}\frac{2}{8} = 7$ such 1 factors.

### A.2.1 HYPERGRAPHS AND *Baranyai's theorem*

Baranyai's theorem in combinatorial mathematics deals with the number of ways one can partition a complete hypergraph into 1-factors. A complete hypergraph $G_r^n$ is a hypergraph containing $n$ vertices in which every subset of $r$ vertices forms a hyperedge. A 1 factor of this graph is a partition of the hypergraph into $\frac{n}{r}$ hyperedges in which each vertex touches exactly one hyperedge. Theorem A.22 gives an informal statement of Baranyai's theorem.

**Theorem A.22** (Baranyai's theorem - Informal). *The $n$ vertices of a hypergraph $G_r^n | n, r \in \mathbb{N}$ such that $r$ divides $n$ can be partitioned into 1-factors in $\binom{n}{r}\frac{r}{n}$ different ways such that each hyperedge in $G_r^n$ appears in exactly one of the partitions (see Fig. 1 for a graphical illustration).*

Theorem A.22 turns out to be useful in getting the almost sure convergence as is stated in Theorem 5.4. Concretely, we will need to reason about the moments of infinite sums of random variables. Specifically, let $\{z^1, ..., z^k\} \in \mathbb{R}^n$ denote independent and normally distributed vectors, let $\psi_{\boldsymbol{\beta}} = \psi(z^1_{\beta_1}, ..., z^k_{\beta_r})$ where $\psi : \mathbb{R}^{kr} \to \mathbb{R}$ is polynomialy bounded and $\forall_{\boldsymbol{\beta}}, \mathbb{E}[\psi_{\boldsymbol{\beta}}] = 0$. Consider the expression:

$$m_q = \mathbb{E}_{z^1, ..., z^k}\Big[\Big(\sum_{\boldsymbol{\beta}=1}^{n} \frac{\psi_{\boldsymbol{\beta}}}{n^{r-0.5}}\Big)^{2q}\Big] \tag{59}$$

**Theorem A.23.** *There exists $C(q) \in [0, \infty)$ such that $\lim_{n \to \infty} m_q \leq C(q)$.*

*Proof.* We can express $m_q$ by breaking the sum $\sum_{\boldsymbol{\beta}=1}^{n}$:

$$m_q = \Big[\Big(\frac{1}{n^{r-0.5}}\sum_{\boldsymbol{\beta}=1}^{n}\psi_{\boldsymbol{\beta}}\mathbb{1}_{\beta_1 \neq \beta_2 \neq ... \neq \beta_r} + \frac{1}{n^{r-0.5}}\sum_{\boldsymbol{\beta}=1}^{n}\psi_{\boldsymbol{\beta}}(1 - \mathbb{1}_{\beta_1 \neq \beta_2 \neq ... \neq \beta_r})\Big)^{2q}\Big] \tag{60}$$

$$\leq A(n) + B(n) \quad \text{where:} \tag{61}$$

$$A = \tilde{C}\Big[\Big(\frac{1}{n^{r-0.5}}\sum_{\boldsymbol{\beta}=1}^{n}\psi_{\boldsymbol{\beta}}\mathbb{1}_{\beta_1 \neq \beta_2 \neq ... \neq \beta_r}\Big)^{2q}\Big] \tag{62}$$

$$B = \tilde{C}\Big[\Big(\frac{1}{n^{r-0.5}}\sum_{\boldsymbol{\beta}=1}^{n}\psi_{\boldsymbol{\beta}}(1 - \mathbb{1}_{\beta_1 \neq \beta_2 \neq ... \neq \beta_r})\Big)^{2q}\Big] \tag{63}$$

where $\tilde{C} \in [0, \infty)$ does not depend on $n$. We now prove the following:

1. $\lim_{n \to \infty} B \to 0$. To see this, notice that there are $n^r - \frac{n!}{(n-r)!}$ non zero terms in the sum $\sum_{\boldsymbol{\beta}=1}^{n} \psi_{\boldsymbol{\beta}}(1 - \mathbb{1}_{\beta_1 \neq \beta_2 \neq \ldots \neq \beta_r})$, hence:

$$B \leq \tilde{C}\Big(\frac{n^r - \frac{n!}{(n-r)!}}{n^{r-0.5}}\Big)^{2q} \max_{\boldsymbol{\beta}} \mathbb{E}\big[(\psi_{\boldsymbol{\beta}})^{2q}\big] \tag{64}$$

Notice that $\lim_{n \to \infty} \frac{n^r - \frac{n!}{(n-r)!}}{n^{r-0.5}} \to 0$, and $\max_{\boldsymbol{\beta}} \mathbb{E}\big[(\psi_{\boldsymbol{\beta}})^{2q}\big]$ is bounded and does not depend on $n$. We can therefore conclude $B$ vanishes as $n \to \infty$.

2. We use Lemma A.21 and Theorem A.22 to show that $\lim_{n \to \infty} A \leq C(q)$ for some $C(q) < \infty$. Let $\Theta = \{\boldsymbol{\beta}^1, \boldsymbol{\beta}^2, \ldots, \boldsymbol{\beta}^{\frac{n!}{(n-r)!}}\}$ denote the set of all possible configurations of $r$ indices $\boldsymbol{\beta}$ such that $\beta_1 \neq \beta_2 \neq \ldots \neq \beta_r$. Let $\{\Theta^i\}_{i=1}^{m}$ denote $m$ sets where each set $\Theta^i$ contains $\frac{n}{r}$ configurations $\Theta^i = \{\boldsymbol{\beta}^{1,i}, \ldots, \boldsymbol{\beta}^{\frac{n}{r},i}\}$ such that:

(a) $\forall_{i;\boldsymbol{\beta},\boldsymbol{\beta}' \in \Theta^i | \boldsymbol{\beta} \neq \boldsymbol{\beta}'}, \; \boldsymbol{\beta} \cap \boldsymbol{\beta}' = \emptyset$

(b) $\forall_{i \neq j;\boldsymbol{\beta} \in \Theta^i;\boldsymbol{\beta}' \in \Theta^j}, \; \boldsymbol{\beta} \neq \boldsymbol{\beta}'$

Finally, let $R$ denote the remaining configurations that do not appear in any set $\{\Theta^i\}$ (i.e $R = \Theta/(\Theta^1 \cup \Theta^2 \cup \ldots \cup \Theta^m)$. We then have:

$$\frac{1}{n^{r-0.5}} \sum_{\boldsymbol{\beta}=1}^{n} \psi_{\boldsymbol{\beta}} \mathbb{1}_{\beta_1 \neq \beta_2 \neq \ldots \neq \beta_r} = \sum_{i=1}^{m} \frac{S^i}{n^{r-1}} + \frac{1}{n^{r-0.5}} \sum_{\boldsymbol{\beta} \in R} \psi_{\boldsymbol{\beta}} \tag{65}$$

$$\text{where} \quad S^i = \frac{1}{\sqrt{n}} \sum_{\boldsymbol{\beta} \in \Theta^i} \psi_{\boldsymbol{\beta}} \tag{66}$$

Note that by construction of the sets $\{\Theta^i\}_{i=1}^{m}$, and since the vectors $z^1, \ldots, z^k$ contain iid coordinates, we can conclude that for all $i$, random variables $\{\psi_{\boldsymbol{\beta}}\}_{\boldsymbol{\beta} \in \Theta^i}$ are independent. Provided we can construct $m = rn^{r-1} - O(n^{r-2})$ sets, then $|R| = O(n^{r-1})$. In that case, then by Lemma A.21:

$$\lim_{n \to \infty} A \leq \lim_{n \to \infty} \tilde{C}\big(\frac{m}{n^{r-1}}\big)^{2q} \mathbb{E}\Big[\big(\sum_{i=1}^{m} \frac{S^i}{m}\big)^{2q}\Big] + \lim_{n \to \infty} \tilde{C}\mathbb{E}\Big[\big(\frac{1}{n^{r-0.5}} \sum_{\boldsymbol{\beta} \in R} \psi_{\boldsymbol{\beta}}\big)^{2q}\Big] \tag{67}$$

$$\leq C(q) + \lim_{n \to \infty} \tilde{C}\big(\frac{|R|}{n^{r-0.5}}\big)^{2q} \max_{\boldsymbol{\beta}} \mathbb{E}[(\psi_{\boldsymbol{\beta}})^{2q}] \tag{68}$$

$$= C(q) + 0 \tag{69}$$

We are then left with proving that we can in fact partition the set $\Theta$ into $\{\Theta^i\}_{i=1}^{m} \cup R$ where $m = rn^{r-1} - O(n^{r-2})$. To show this, we define a complete hypergraph $G_r^n$ with $n$ vertices in which every vertex corresponds to an integer in $\{1, 2, \ldots, n\}$. We can think of a hyperedge in $G_r^n$ as an edge connecting $r$ integers without ordering, hence the set of all hyperedges in $G_r^n$ has cardinality $\frac{1}{r!}|\Theta|$ (this is since ordering matters in $\Theta$). By Theorem A.22, we can partition the vertices in $G_r^n$ into $\frac{n}{r}$ hyperedges (sets of $r$ unique integers with no ordering) in $\binom{n}{r}\frac{r}{n}$ different ways, where each hyperedge appears in a single partition. For each partition $i$, we can assign $\Theta^i$ where any $\boldsymbol{\beta} \in \Theta^i$ corresponds to a single hyperedge in partition $i$, with the ordering of the vertices decided arbitrarily. Notice that for each partition $\Theta^i$, we can construct $(r! - 1)$ additional partitions by reordering $\boldsymbol{\beta}$ in any $\boldsymbol{\beta} \in \Theta^i$ Therefore, the total number of valid partition is given by $\binom{n}{r}\frac{r}{n}r! = rn^{r-1} - O(n^{r-2})$, proving the theorem.

$\square$

## A.3 BASE CASE

WLOG, we start with an initial corset of Gaussian iid vectors $\boldsymbol{x} = \{x^1, ..., x^k\}$ (which are regular), an initial vanishing set of vanishing vectors $\hat{\boldsymbol{x}} = \{\hat{x}^1, ..., \hat{x}^{k'}\}$ and a set of vanishing scalars $\Theta = \{\theta_1, ..., \theta_l\}$. Note that **ReWrite**(1) and **StableRank**(1) trivially hold. We proceed to prove **Dichotomy**(1),**TensorMoment**(1) and **ConvRate**(1).

We define the functions $\psi : \mathbb{R}^k \to \mathbb{R}, \bar{\psi} : \mathbb{R}^{(r+1)k} \to \mathbb{R}, \bar{\psi} : \mathbb{R}^k \to \mathbb{R}$ using the pseudo lipschitz function $\psi : \mathbb{R}^{(r+1)k} \to \mathbb{R}$ and vectors $\boldsymbol{x}$:

$$\psi(y) \overset{\text{def}}{=} \mathbb{E}_{Z_1^{\boldsymbol{x}}, Z_2^{\boldsymbol{x}}, ..., Z_r^{\boldsymbol{x}}} \Big[ \psi\big(y; Z_1^{\boldsymbol{x}}, Z_2^{\boldsymbol{x}}, ..., Z_r^{\boldsymbol{x}}\big) \Big] \tag{70}$$

$$\bar{\psi}(y; \boldsymbol{x_\beta}) \overset{\text{def}}{=} \psi(y, \boldsymbol{x_\beta}) - \psi(y) \tag{71}$$

$$\bar{\psi}(y) \overset{\text{def}}{=} \frac{1}{n^r} \sum_{\boldsymbol{\beta}=1}^{n} \bar{\psi}(y; \boldsymbol{x_\beta}) \tag{72}$$

$$\bar{h}_\alpha \overset{\text{def}}{=} \psi(\boldsymbol{x_\alpha}) \tag{73}$$

$$\Delta \bar{h}_\alpha \overset{\text{def}}{=} \bar{\psi}(\boldsymbol{x_\alpha}) \tag{74}$$

Note that $\bar{\psi}(y)$ is a random function that depends on the vectors $\boldsymbol{x}$.

**Theorem A.24.** $\Delta \bar{h}$ *is a vanishing vector.*

*Proof.* From Lemma A.20, it suffices to show that for every $p > 0$, there exists $t \in \mathbb{N}$ such that $\mathbb{E} \frac{\left( \sum_{\alpha=1}^{n} \bar{\psi}(\boldsymbol{x_\alpha})^2 \right)^t}{n^{tp}} \le cn^{-1-\lambda}$ for some $c, \lambda > 0$ (which may depend on $p$). Fix $p > 0$, and choose $t = \frac{1+\lambda}{p}$, and let $q = \lceil t \rceil$. Then by *Jensen's inequality*:

$$\mathbb{E} \Big( \frac{\left( \sum_{\alpha=1}^{n} \bar{\psi}(\boldsymbol{x_\alpha})^2 \right)^t}{n^p} \Big) \le \frac{1}{n^{1+\lambda}} \mathbb{E} \Big[ \Big( \sum_{\alpha=1}^{n} \bar{\psi}(\boldsymbol{x_\alpha})^2 \Big)^q \Big] \tag{75}$$

$$= \frac{1}{n^{1+\lambda}} \mathbb{E} \Big[ \Big( \sum_{\alpha=1}^{n} \Big( \sum_{\boldsymbol{\beta}=1}^{n} \frac{\bar{\psi}(\boldsymbol{x_\alpha}, \boldsymbol{x_\beta})}{n^r} \Big)^2 \Big)^q \Big] \tag{76}$$

$$\le \frac{1}{n^{1+\lambda}} \mathbb{E} \Big[ \Big( \sum_{\boldsymbol{\beta}=1}^{n} \frac{\bar{\psi}(\boldsymbol{x_1}, \boldsymbol{x_\beta})}{n^{r-0.5}} \Big)^{2q} \Big] \tag{77}$$

We are now left with the task of proving that $\mathbb{E} \Big[ \Big( \sum_{\boldsymbol{\beta}=1}^{n} \frac{\bar{\psi}(\boldsymbol{x_1}, \boldsymbol{x_\beta})}{n^{r-0.5}} \Big)^{2q} \Big]$ is finite for any (fixed) integer $q$ and for $n \to \infty$. Firstly, we may express the over all indices $\sum_{\boldsymbol{\beta}=1}^{n}$ as $\sum_{\boldsymbol{\beta}=2}^{n} + \sum_{\boldsymbol{\beta}|\exists j:\beta_j=1}^{n}$. That is, we first sum over all the indices where each one is bigger than 1, and then sum over all indices where at least one of them is 1. Then we have:

$$\mathbb{E} \Big[ \Big( \sum_{\boldsymbol{\beta}=1}^{n} \frac{\bar{\psi}(\boldsymbol{x_1}, \boldsymbol{x_\beta})}{n^{r-0.5}} \Big)^{2q} \Big] = \mathbb{E} \Big[ \Big( \sum_{\boldsymbol{\beta}=2}^{n} \frac{\bar{\psi}(\boldsymbol{x_1}, \boldsymbol{x_\beta})}{n^{r-0.5}} + \sum_{\boldsymbol{\beta}|\exists j:\beta_j=1}^{n} \frac{\bar{\psi}(\boldsymbol{x_1}, \boldsymbol{x_\beta})}{n^{r-0.5}} \Big)^{2q} \Big] \tag{78}$$

$$\le \tilde{C} \mathbb{E} \Big[ \Big( \sum_{\boldsymbol{\beta}=2}^{n} \frac{\bar{\psi}(\boldsymbol{x_1}, \boldsymbol{x_\beta})}{n^{r-0.5}} \Big)^{2q} \Big] + \tilde{C} \mathbb{E} \Big[ \Big( \sum_{\boldsymbol{\beta}|\exists j:\beta_j=1}^{n} \frac{\bar{\psi}(\boldsymbol{x_1}, \boldsymbol{x_\beta})}{n^{r-0.5}} \Big)^{2q} \Big] \tag{79}$$

where $\tilde{C} \le \infty$ does not depend on $n$. we now make the following observations:

**Claim A.3.** *There exists a constant $C < \infty$ that is independent of $n$ such that $\forall_{n,r}, \mathbb{E} \Big[ \Big( \sum_{\boldsymbol{\beta}|\exists j:\beta_j=1}^{n} \frac{\bar{\psi}(\boldsymbol{x_1}, \boldsymbol{x_\beta})}{n^{r-0.5}} \Big)^{2q} \Big] \le C$. To see this, note that the summation (i.e $\sum_{\boldsymbol{\beta}|\exists j:\beta_j=1}^{n}$) effectively sums over $n^r - (n-1)^r \sim O(n^{r-1})$ terms. Since $\bar{\psi}$ is a centered pseudo lipschitz function of normally distributed variables, we can bound the second expectation:*

$$\forall_{n,r}, \mathbb{E} \Big[ \Big( \sum_{\boldsymbol{\beta}|\exists j:\beta_j=1}^{n} \frac{\bar{\psi}(\boldsymbol{x_1}, \boldsymbol{x_\beta})}{n^{r-0.5}} \Big)^{2q} \Big] \le \frac{\tilde{C}}{n^q} \max_{\boldsymbol{\beta}} \mathbb{E} \big[ \bar{\psi}(\boldsymbol{x_1}, \boldsymbol{x_\beta})^{2q} \big] \le C \tag{80}$$

*for some $\tilde{C} < \infty$ that do not depend on $n$.*

**Claim A.4.** *There exists a constant $C < \infty$ that is independent of $n$ such that $\forall n, r, \quad \mathbb{E}\left[\left(\sum_{\boldsymbol{\beta}=2}^{n} \frac{\bar{\psi}(\boldsymbol{x}_1, \boldsymbol{x}_{\boldsymbol{\beta}})}{n^{r-0.5}}\right)^{2q}\right] \leq C$. Note that since the summation over the indices $\boldsymbol{\beta}$ do not include the first index, the random variable $\boldsymbol{x}_{\boldsymbol{\beta}}$ can be treated as independent from $\boldsymbol{x}_1$. WLOG we can then bound the following term instead:*

$$\forall n, r, \quad \mathbb{E}\left[\left(\sum_{\boldsymbol{\beta}=1}^{n} \frac{\bar{\psi}(y, \boldsymbol{x}_{\boldsymbol{\beta}})}{n^{r-0.5}}\right)^{2q}\right] \leq C \tag{81}$$

*For some random variable $y$ independent of $\boldsymbol{x}_{\boldsymbol{\beta}}$ for all values of $\boldsymbol{\beta}$, with the same dimensions as $\boldsymbol{x}_1$. We can now condition on $y$, and apply Theorem A.23 to complete the proof.*

$\square$

Now, we can express $h = \bar{h} + \Delta h + \Delta \bar{h}$, where $\Delta h, \Delta \bar{h}$ are vanishing vectors. Invoking Claim A.2, it is enough to prove the base case holds for $\bar{h}$ alone:

1. **TensorMoment**$(1)$ is immediate from the law of large numbers given that $\boldsymbol{x}$ are iid gaussians. Namely:

$$\frac{1}{n} \sum_{\alpha=1}^{n} \psi(\boldsymbol{x}_\alpha) \overset{\text{a.s}}{=} \mathbb{E}_{Z^{\boldsymbol{x}}} \psi(Z^{\boldsymbol{x}}). \tag{82}$$

2. **Dichotomy**$(1)$ holds since $\psi$ is a smooth, pseudo lipschitz (given by gaussian averaging of $\psi$) function. From **TensorMoment**$(1)$, for any $p > 0$:

$$\frac{1}{n} \lim_{n \to \infty} \|\bar{h}\|_p^p = \lim_{n \to \infty} \frac{\sum_{\alpha=1}^{n} |\bar{h}_\alpha|^p}{n} \overset{\text{a.s}}{\to} \mathbb{E}_{Z^{\boldsymbol{x}}}\left[|\psi(Z^{\boldsymbol{x}})|^p\right] = \int_{Z^{\boldsymbol{x}}} |\psi(Z^{\boldsymbol{x}})|^p d\mathcal{N}(Z^{\boldsymbol{x}}) dZ^{\boldsymbol{x}} \tag{83}$$

   where $\mathcal{N}(Z^{\boldsymbol{x}})$ is the gaussian measure. Therefore, if $\frac{1}{n} \lim_{n \to \infty} \|\bar{h}\|_p^p = 0$ for some $p > 0$, then $\psi$ is identically zero. In that case, we can write $h = \boldsymbol{0} + \Delta h + \Delta \bar{h}$, which is a vanishing vector. If $\psi$ is not identically zero, then $h$ is a regular vector from **TensorMoment**$(1)$, proving **Dichotomy**$(1)$.

3. **ConvRate**$(1)$ holds since for all $p > 0$:

$$\lim_{n \to \infty} \frac{1}{n^{p-1}}\left(\frac{\sum_{\alpha=1}^{n}(\bar{h}_\alpha - \mathring{\theta})}{n}\right)^2 = \lim_{n \to \infty} \frac{1}{n^p}\left(\frac{\sum_{\alpha=1}^{n}(\bar{h}_\alpha - \mathring{\theta})}{\sqrt{n}}\right)^2 \overset{\text{a.s}}{=} 0 \tag{84}$$

   which holds from Theorem A.24 and assigning $r = 0$.

We have therefor concluded the base case.

## A.4 INDUCTION STEP

We prove the induction step assuming **IH**$(m)$ holds. Namely, we must show that **IH**$(m) \to$ **IH**$(m+1)$. Assume a new vector is introduced via MATMUL, namely $W\nu$ where $\nu$ is given by TENSOR operation of corset $\boldsymbol{x}$, vanset $\hat{\boldsymbol{x}}$ and vanishing scalars $\Theta$. By **Dichotomy**$(m)$, we may express $\nu = \bar{\nu} + \Delta\nu + \Delta\bar{\nu}$ where $\Delta\nu + \Delta\bar{\nu}$ is a vanishing vector, and $\bar{\nu}$ is either regular or vanishing.

### A.4.1 IH$(m) \to$ REWRITE$(m+1)$ + STABLERANK$(m+1)$

We can express $W\nu = W\bar{\nu} + W(\Delta\nu + \Delta\bar{\nu})$, and point to the following fact:

**Claim A.5.** *If $\bar{\nu}$ is a regular vector, then $W\bar{\nu}$ is a regular vector. Moreover, for any vanishing vector $\delta \in \mathbb{R}^n$, $W\delta$ is a vanishing vector.*

*Proof.* If $\delta$ is vanishing then $W\delta$ is vanishing: This is true since $W$ is a gaussian matrix with a uniformally bounded (in $n$) operator norm. The first part Claim A.5 holds since $\bar{\nu}$ depends only on vectors from $\boldsymbol{x}$, for which the set $\boldsymbol{x}_W$ has a stable rank from **StableRank**$(m)$. We can therefore use the gaussian conditioning trick (conditioning on all vectors in $\boldsymbol{x}$ and $\boldsymbol{x}_W$). $\qquad\square$

We can now expand the vanset with $\hat{\boldsymbol{x}} := \hat{\boldsymbol{x}} \cup W(\Delta\nu + \Delta\bar{\nu})$, and proceed by casework:

1. If $\bar{\nu}$ is vanishing then we expand $\hat{\boldsymbol{x}} := \hat{\boldsymbol{x}} \cup W\bar{\nu}$. In that case $\boldsymbol{x}$ remains unchanged and **StableRank**$(m+1)$ trivially holds.

2. If $\bar{\nu}$ is regular then we expand $\boldsymbol{x} := \boldsymbol{x} \cup W\bar{\nu}$, and we get **StableRank**$(m+1)$ using Theorem A.25.

**Theorem A.25.** *Let $x^1, ..., x^k \in \boldsymbol{x}_W$. If $Z^x = \alpha_1 Z^{x^1} + \alpha_2 Z^{x^2} + ... + Z^{x^k}$, then almost surely, for large enough $n$, $x = \alpha_1 x^1 + \alpha_2 x^2 + ... + \alpha_k x^k$.*

*Proof.* The set $\boldsymbol{x}$ is constructed as a standard NETSOR⊤program (without scalars), and we may immediately apply theorem 6.3 in Yang (2020b). $\qquad\square$

### A.4.2    **IH**$(m)$ + **REWRITE**$(m+1)$ + **STABLERANK**$(m+1)$ → **IH**$(m+1)$

We are left with proving **Dichotomy**$(m+1)$, **TensorMoment**$(m+1)$ and **ConvRate**$(m+1)$. Note that if $\bar{\nu}$ is a vanishing vector, the set $\boldsymbol{x}$ remains unchanged, and we immediately get **IH**$(m+1)$ by Claim A.2. Hence we proceed assuming $\bar{\nu}$ (hence $W\bar{\nu}$ is a regular vector.

**Getting Dichotomy**$(m+1)$    Assume a new vector is introduced in the program via a TENSOR operation:

$$h_\alpha = \frac{1}{n^r} \sum_{\boldsymbol{\beta}} \psi(\boldsymbol{x}_{\alpha,\boldsymbol{\beta}}, (W\bar{\nu})_{\alpha,\boldsymbol{\beta}}; \hat{\boldsymbol{x}}_{\alpha,\boldsymbol{\beta}}; \Theta) \tag{85}$$

Where we made explicit the inclusion to $\boldsymbol{x}$ of the new vector $W\bar{\nu}$. Let $h^0, \bar{h}, \Delta h, \Delta\bar{h}$ be defined as in Eq. (13). Note that $\Delta h$ is vanishing by Claim A.1, which holds generally. We next prove $\Delta\bar{h} = h^0 - \bar{h}$ is a vanishing vector, where (using $\psi(-) \equiv \psi(-; \boldsymbol{0}; \mathring{\boldsymbol{\Theta}})$ to ease notational burden):

$$\Delta\bar{h}_\alpha = \frac{1}{n^r} \sum_{\boldsymbol{\beta}=1}^{n} \psi(\boldsymbol{x}_{\alpha,\boldsymbol{\beta}}, (W\bar{\nu})_{\alpha,\boldsymbol{\beta}}) - \mathbb{E}_{\boldsymbol{Z}_1^{\boldsymbol{x}}, Z_1^{W\bar{\nu}}, ..., \boldsymbol{Z}_r^{\boldsymbol{x}}, Z_r^{W\bar{\nu}}} \left[ \psi(\boldsymbol{x}_\alpha, (W\bar{\nu})_\alpha, \boldsymbol{Z}_1^{\boldsymbol{x}}, Z_1^{W\bar{\nu}}, ..., \boldsymbol{Z}_r^{\boldsymbol{x}}, Z_r^{W\bar{\nu}}) \right]$$

$$\tag{86}$$

The key insight to proving $\Delta\bar{h} = h^0 - \bar{h}$ is indeed vanishing is that both $h^0, \bar{h}$ can be written as a sum of a shared regular vector, and a vanishing vector (i.e we can express $h^0 = \mu + \delta^1, \bar{h} = \mu + \delta^2$ where $\mu$ is regular, and $\delta^1, \delta^2$ are vanishing). Their difference $h^0 - \bar{h} = \delta^1 - \delta^2$ is therefore the difference of two vanishing vectors, which is itself vanishing. To show this, we can make explicit the distribution of $W\bar{\nu}$ by using the gaussian conditioning trick (see Appendix A.5). Denote by $X, Y \in \mathbb{R}^{n \times r}, U, V \in \mathbb{R}^{n \times s}$ the matrices with $\{x^i\} \in \boldsymbol{x}, \{y^i\} \in \boldsymbol{x}_W, \{u^i\} \in \boldsymbol{x}, \{v^i\} \in \boldsymbol{x}_{W^\top}$ as columns respectively, representing previously generated vectors in the program, such that $X = WY, U = W^\top V$. Using the Gaussian conditioning trick (conditioning on all the vectors in $\boldsymbol{x}$), $g = W\bar{\nu}$ is distributed as:

$$W\bar{\nu} \stackrel{d}{=} \sum_i d_i x^i + \sum_i e_i v^i + \sigma\Pi_V^\perp z \tag{87}$$

where $d \to \mathring{d}, e \to \mathring{e}, \sigma \to \mathring{\sigma}$, and $z \sim \mathcal{N}(0, I_n)$. Define:

$$a = \sum_i \mathring{d}_i x^i + \sum_i \mathring{e}_i v^i + \mathring{\sigma} z \tag{88}$$

$$b = g - a = \sum_i (d_i - \mathring{d}_i) x^i + \sum_i (e_i - \mathring{e}_i) v^i + (\sigma\Pi_V^\perp - \mathring{\sigma}) z \tag{89}$$

We now note that:

- $\forall_i$, $(d_i - \mathring{d}_i)x^i$ is a vanishing vector since $x^i$ is regular ($x \in \boldsymbol{x}_W$), and $(d_i - \mathring{d}_i)$ is vanishing by the induction hypothesis, and Lemma A.17.

- $\forall_i$, $(e_i - \mathring{e}_i)v^i$ is a vanishing vector since $v^i$ is regular ($v \in \boldsymbol{x}_{W^\top}$), and $(e_i - \mathring{e}_i)$ is vanishing by the induction hypothesis, and Lemma A.17.

- $(\sigma \Pi_V^\perp - \mathring{\sigma})z$ is a vanishing vector. To see this, note that:

$$(\sigma \Pi_V^\perp - \mathring{\sigma})z = (\mathring{\sigma} - \sigma)z + V(VV^\top)^\dagger V^\top z \tag{90}$$

$(\mathring{\sigma} - \sigma)z$ is vanishing due to the induction hypothesis and Lemma A.17. $V(VV^\top)^\dagger V^\top z$ is vanishing as well. To see this, note that:

$$\forall_{p>0}, \quad \frac{\|V(VV^\top)^\dagger V^\top z\|^2}{n^p} = \frac{\|\frac{1}{n} V (\frac{VV^\top}{n})^\dagger V^\top z\|^2}{n^p} \tag{91}$$

$$= \frac{z^\top V (\frac{VV^\top}{n})^\dagger V^\top z}{n^{p+1}} = \frac{1}{n^p} \frac{z^\top V}{\sqrt{n}} \left(\frac{VV^\top}{n}\right)^\dagger \frac{V^\top z}{\sqrt{n}} \tag{92}$$

By the induction hypothesis $(\frac{VV^\top}{n})^\dagger$ converges almost surely ($V$ has stable rank). Then it is enough to show that $\forall_{i,p>0}$, $\lim_{n\to\infty} \frac{1}{n^{\frac{p}{2}}} \frac{z^\top v^i}{\sqrt{n}} \overset{\text{a.s}}{\to} 0$. From **TensorMoment**$(m)$ and **ConvRate**$(m)$:

$$\forall_{i,p>0}, \quad \lim_{n\to\infty} \frac{1}{n^{\frac{p}{2}}} \frac{z^\top v^i}{\sqrt{n}} = \lim_{n\to\infty} \sqrt{n^{1-p}(\frac{z^\top v^i}{n})^2} \tag{93}$$

$$= \sqrt{\lim_{n\to\infty} n^{1-p}(\frac{z^\top v^i}{n})^2} \overset{\text{a.s}}{\to} 0 \tag{94}$$

We therefore conclude that $b = g - a$ is vanishing. We have:

$$h^0 = \frac{1}{n^r} \sum_{\boldsymbol{\beta}=1}^n \psi(\boldsymbol{x}_{\alpha,\boldsymbol{\beta}}, g_{\alpha,\boldsymbol{\beta}}) = \frac{1}{n^r} \sum_{\boldsymbol{\beta}=1}^n \psi(\boldsymbol{x}_{\alpha,\boldsymbol{\beta}}, a_{\alpha,\boldsymbol{\beta}}; b_{\alpha,\boldsymbol{\beta}}) = A(m) + B(m) \tag{95}$$

$$\text{where} \quad A(m) = \frac{1}{n^r} \sum_{\boldsymbol{\beta}=1}^n \psi(\boldsymbol{x}_{\alpha,\boldsymbol{\beta}}, (\sum_i \mathring{d}_i x^i + \sum_i \mathring{e}_i v^i + \mathring{\sigma} z)_{\alpha,\boldsymbol{\beta}}) \tag{96}$$

$$B(m) = \frac{1}{n^r} \sum_{\boldsymbol{\beta}=1}^n \left[ \psi(\boldsymbol{x}_{\alpha,\boldsymbol{\beta}}, a_{\alpha,\boldsymbol{\beta}}; b_{\alpha,\boldsymbol{\beta}}) - \psi(\boldsymbol{x}_{\alpha,\boldsymbol{\beta}}, (\sum_i \mathring{d}_i x^i + \sum_i \mathring{e}_i v^i + \mathring{\sigma} z)_{\alpha,\boldsymbol{\beta}}) \right] \tag{97}$$

From Claim A.1, $B(m)$ is a vanishing vector. Furthermore, $a$ is a deterministic function of previous vectors in $\boldsymbol{x}$, and an iid gaussian noise vector $z$.
We can now recursively expand $A(m) = A(m-1) + B(m-1)$ where $B(m-1)$ is a vanishing vector, until we are left with $A(1)$ (i.e $A(m) = A(1) + \sum_{m'=1}^{m-1} B(m')$, where $\{B(m')\}_{m'}$ are all vanishing vectors). Note that $A(1)$ can be expressed as a pseudo lipschitz function of normally distributed vectors, with coordinate distributions given by $Z^g, Z^{x^1}, ..., Z^{x^{|\boldsymbol{x}|}}$. We can apply the same decomposition to $\bar{h}$ and get $\bar{h} = \bar{A}(1) + \sum_{m'=1}^{m-1} \bar{B}(m')$. We have that:

$$\Delta \bar{h} = h^0 - \bar{h} = A(1) - \bar{A}(1) + \sum_{m'=1}^{m-1} (B(m') - \bar{B}(m')) \tag{98}$$

Finally, it is easy to see that $A(1) = \bar{A}(1)$, and hence we may invoke the base case (in particular Theorem A.24) and conclude that $A(m) - \bar{A}(m)$, and by extension $\Delta \bar{h}$ are vanishing.

**Getting TensorMoment**$(m+1)$ **and ConvRate**$(m+1)$    These are immediate since we can express $h = \bar{A}(1) + \delta$ where $\bar{A}(1)$ is a function of gaussian vectors, and $\delta$ is a vanishing vector, and by Claim A.2 invoke the base case on $\bar{A}(1)$.

### A.5 GAUSSIAN CONDITIONING

Let $g = Wh$ where $g, h \in \mathbb{R}^n$ are vectors in a NETSOR⊤ program.

Denote by $X, Y \in \mathbb{R}^{n \times r}, U, V \in \mathbb{R}^{n \times s}$ the matrices with $\{x^i\}, \{y^i\}, \{u^i\}, \{v^i\}$ as columns respectively, representing previously generated vectors in the program, such that $X = WY, U = W^\top V$. Using the Gaussian conditioning trick (conditioning on all the vectors in $\boldsymbol{x}$), $g = Wh$ is distributed as:

$$g \overset{d}{=} (E + \Pi_V^\perp \tilde{W} \Pi_Y^\perp)\nu^0 = A + B$$

where we have defined $A = E\nu^0, B = \Pi_V^\perp \tilde{W} \Pi_Y^\perp h$, $\Pi_V, \Pi_Y$ are projection matrices, and $\tilde{W}$ is a fresh iid sample of $W$, and:

$$E = XY^+ + V^{+\top}U^\top - V^{+\top}U^\top YY^+$$

Rewriting the conditional distribution of $g$, we get:

$$g \overset{\mathrm{D}}{=} \Theta + \sigma \Pi_V^\perp z \tag{99}$$

$$\text{with } \Theta \overset{\text{def}}{=} Eh \in \mathbb{R}^n, \quad \sigma \overset{\text{def}}{=} \sigma_A \sqrt{\frac{\|\Pi_Y^\perp h\|^2}{n}} \in \mathbb{R} \tag{100}$$

Moreover, $\sigma$ converges to a deterministic limit $\mathring{\sigma}$, and $\Theta$ can be written as:

$$\Theta = X(\mathring{d} + \hat{\epsilon}) + V(\mathring{e} + \check{\epsilon}) \tag{101}$$

where $\hat{\epsilon} \in \mathbb{R}^r, \check{\epsilon} \in \mathbb{R}^s$ are vanishing vectors, and $\mathring{d} \in \mathbb{R}^r, \mathring{e} \in \mathbb{R}^s$ are deterministic vectors.

□

## B PROOFS OF THEOREM 4.1 AND THEOREM 4.2

Now that we have proven Theorem 5.4, we prove Theorem 4.1 and Theorem 4.2 by writing out the TPs which implements the training process, and apply the master theorem. To accomplish this, we start by expressing the explicit computation done at each training step at finite width, implement it as a set of TP instructions, convert it to infinite-width computation according to Definition 5.3 and apply the master theorem.

### B.1 THE TENSOR PROGRAM FOR THEOREM 4.1

In the next section, we construct the Tensor Program that encodes the training of an L-hidden layer MLP as in Eq. (1) under the ANKT parametrization. Here we first describe the initial matrices, vectors, and scalars of the program, along with necessary notations.

**Initial Matrices, Vectors and Scalars**  We first define the initial set of matrices $\mathcal{W}$, vectors $\mathcal{V}$ and scalars $\mathcal{C}$:

- Initial matrices $\{w^l\}_{l=2}^L \in \mathbb{R}^{n \times n}$ are all sampled iid from $\mathcal{N}(0,1)$. We set $\mathcal{W} = W^2 \cup W^3 \cup ... \cup W^{L+1}$.

- The initial vectors $\mathcal{V}$ are given by the first layer $h^1(\xi)$ for all inputs, and the last weight vector $w^{L+1} \in \mathbb{R}^n$, all samples iid from $\mathcal{N}(0,1)$.

- Initial scalar $\mathcal{C} = \{\frac{1}{\sqrt{n}}\}$.

**Notations**  We use $:=$ to more clearly denote assignment happening in the program, as opposed to mathematical equality. We will use the notation $\text{TENSOR}(y^1, .., y^k; \Theta), \text{TENSORMOMENT}(y^1, .., y^k; \Theta)$ to denote an arbitrary implementation of these instructions give vectors $y^1, ..., y^k$ and scalars $\Theta$.

**Initial Forward Pass** Starting with our initial vectors $h^1(\xi) := w^1\xi$ , we compute all $\{x^l(\xi)\}_{l=1}^L, \{h^l(\xi)\}_{l=2}^L$ using TENSOR and MATMUL instructions:

$$\forall_{1 \le l \le L}, \; x^l(\xi) := \psi(h^l) \text{ for } \psi(y) \stackrel{\text{def}}{=} \phi(y) \tag{102}$$

$$\forall_{1 < l \le L}, \; h^l(\xi) := W^l x^{l-1}(\xi) \tag{103}$$

The initial outputs are given by $f(\xi) = \frac{w^{L+1\top} x^L(\xi)}{\sqrt{n}}$ since TENSORMOMENT only allows division by $n^r$ for integer $r$, rather than $\sqrt{n}$. However recall that in Theorem 4.1 we assume WLOG that the outputs for any input $\xi$ is fixed to $f(\xi) = g(\xi)$. Let $X$ denote a matrix composed of all $x^L(\xi)$ as columns. Denote by $e$ the event that $f(\xi) = 0$ for all inputs. Then, using gaussian conditioning, the conditional distribution of $w_e^{L+1}$ given $e$ is:

$$w_e^{L+1} \stackrel{D}{=} \Pi \tilde{w}^{L+1} \tag{104}$$

where $\tilde{w}^{L+1}$ is an independent sample of $w^{L+1}$, and $\Pi = I - \frac{1}{n} X(\frac{X^\top X}{n})^\dagger X^\top$ and $\bullet^\dagger$ is the pseudo-inverse of $\bullet$. Then, we can write:

$$w_e^{L+1} \stackrel{D}{=} \tilde{w}^{L+1} - X\left(\frac{X^\top X}{n}\right)^\dagger \frac{X^\top \tilde{w}^{L+1}}{n} \tag{105}$$

By the master theorem $\frac{X^\top X}{n} \stackrel{\text{a.s}}{\to} \gamma$, $(\frac{X^\top X}{n})^\dagger \stackrel{\text{a.s}}{\to} \gamma^\dagger$ and $\frac{X^\top \tilde{w}^{L+1}}{n} \stackrel{\text{a.s}}{\to} 0$. Hence, after conditioning on $f(\xi) = 0$, the distribution of $w_e^{L+1}$ is still identical to that of $w^{L+1}$ at the limit. At this point we can just implement $w^{L+1}$ in the program with $\tilde{w}^{L+1} - X(\frac{X^\top X}{n})^\dagger \frac{X^\top \tilde{w}^{L+1}}{n}$ which can be implemented with TENSOR and TENSORMOMENT instructions. We now have a program that encodes the initial forward pass of the MLP conditioned on $f(\xi) = 0$ for all $\xi$.

**Initial Backward Pass and Loss Derivatives** For any input sample $\tilde{\xi}$, we can implement $d\tilde{h}^l$ using TENSOR:

$$d\tilde{h}^L := \text{TENSOR}(h^L(\xi), w^{L+1}) \text{ for } \text{TENSOR}(y^1, y^2) \stackrel{\text{def}}{=} \phi'(y^1) \odot y^2 \tag{106}$$

Then, for all $1 \le l < L$, using MATMUL and TENSOR:

$$d\tilde{h}^l := \text{TENSOR}(W^{l+1\top} d\tilde{h}^{l+1}, \tilde{h}^l) \text{ for } \text{TENSOR}(y^1, y^2) \stackrel{\text{def}}{=} \phi'(y^1) \odot y^2 \tag{107}$$

The initial loss derivatives $\mathcal{L}'(\xi)$ are all deterministic scalars since we have conditioned on the initial outputs $f(\xi)$.

**Forward and Backward Passes at Any $t$** The forward and backward and loss computation for any $t$ are given by:

$$\tilde{h}_t^l = \left(W^l + \frac{1}{\sqrt{n}} \sum_{t'=0}^{t-1} \Delta w_{t'}^l\right) \tilde{x}_t^{l-1} \tag{108}$$

$$d\tilde{h}_t^L = \phi'(\tilde{h}_t^L) \odot w^{L+1} \tag{109}$$

$$\forall_{1 \le l < L}, \; d\tilde{h}_t^l = \left[\left(W^{l+1} + \frac{1}{\sqrt{n}} \sum_{t'=0}^{t-1} \Delta w_{t'}^{l+1}\right)^\top d\tilde{h}_t^{l+1}\right] \odot \phi'(\tilde{h}_t^l) \tag{110}$$

with the weight updates given by:

$$\Delta w_t^l = -\frac{1}{n} Q(dh_t^l x_t^{l-1\top} \mathcal{L}_t') \tag{111}$$

which can all be implemented with TENSOR and Matmul operations as before.

**Adaptive Update at Time** $t$    Using Eq. (2) and $w_t^l = w^l + \sum_{t'=0}^{t} \Delta w_{t'}^l$ (recall $w^l = \sqrt{n}W^l$), we have that:

$$\delta \tilde{h}_t^2 = \Delta w_t^2 \tilde{h}^1 \tag{112}$$

$$\forall_{2<l\leq L}, \quad \delta \tilde{h}_t^l = \Delta w_t^l \tilde{x}_t^{l-1} + \frac{1}{\sqrt{n}}\Big(w^l + \sum_{t'=0}^{t} \Delta w_{t'}^l\Big)\delta \tilde{x}_t^{l-1} + \frac{1}{\sqrt{n}}\Delta w_t^l \delta \tilde{x}_t^{l-1} \tag{113}$$

$$= -\frac{1}{n}Q(dh_t^l x_t^{l-1\top}\mathcal{L}_t')\tilde{x}_t^{l-1} + \Big(W^l - \frac{1}{n\sqrt{n}}\sum_{t'=0}^{t} Q(dh_{t'}^l x_{t'}^{l-1\top}\mathcal{L}_{t'}')\Big)\delta \tilde{x}_t^{l-1} \tag{114}$$

$$- \frac{1}{n\sqrt{n}}Q(dh_t^l x_t^{l-1\top}\mathcal{L}_t')\delta \tilde{x}_t^{l-1} \tag{115}$$

$$\delta \tilde{x}_t^l = \sqrt{n}\phi(\tilde{h}_t^l + \frac{\delta \tilde{h}_t^l}{\sqrt{n}}) - \sqrt{n}\phi(\tilde{h}_t^l) \tag{116}$$

which can be implemented using TENSOR:

$$\delta \tilde{h}_t^2 := \text{TENSOR}(dh_t^2, x_t^1, \tilde{x}_t^1; \mathcal{L}_t') \text{ for } \text{TENSOR}(y^1, y^2, y^3; \theta) \stackrel{\text{def}}{=} -\frac{1}{n}Q(y^1 y^{2\top}\theta)y^3 \tag{117}$$

$$\delta \tilde{x}_t^2 := \text{TENSOR}(\tilde{h}_t^2, \delta \tilde{h}_t^2; \frac{1}{\sqrt{n}}) \text{ for } \text{TENSOR}(y^1, y^2; \theta) \stackrel{\text{def}}{=} \begin{cases} \frac{1}{\theta}\phi(y^1 + \theta y^2) - \frac{1}{\theta}\phi(y^1) & \theta > 0 \\ \phi'(y^1) \odot y^2 & \theta = 0 \end{cases} \tag{118}$$

and similarly for any layer $2 < l \leq L$.

The (pre)activations at any step $t$ can be implemented as follows using TENSOR:

$$\forall_{1<l\leq L}, \tilde{h}_{t+1}^l := \text{TENSOR}(\tilde{h}_t^l, \delta \tilde{h}_t; \frac{1}{\sqrt{n}}) \text{ for } \text{TENSOR}(y^1, y^2; \theta) \stackrel{\text{def}}{=} y^1 + \theta y^2 \tag{119}$$

$$\forall_{1<l\leq L}, \tilde{x}_{t+1}^l := \text{TENSOR}(\tilde{x}_t^l, \delta \tilde{x}_t; \frac{1}{\sqrt{n}}) \text{ for } \text{TENSOR}(y^1, y^2; \theta) \stackrel{\text{def}}{=} y^1 + \theta y^2 \tag{120}$$

**Output Updates**    The function updates can be implemented using TENSORMOMENT:

$$\Delta \tilde{f}_t := \text{TENSORMOMENT}(w^{L+1}, \delta \tilde{x}_t^L) \text{ for } \text{TENSORMOMENT}(y^1, y^2) \stackrel{\text{def}}{=} \frac{1}{n}\sum_{\alpha=1}^{n} y_\alpha^1 y_\alpha^2 \tag{121}$$

The loss derivatives can be implemented using using TENSORMOMENT:

$$\mathcal{L}_t' := \text{TENSORMOMENT}(; f_t) \text{ for } \text{TENSORMOMENT}(; \theta) \stackrel{\text{def}}{=} \frac{1}{n}\sum_{\alpha=1}^{n} \mathcal{L}'(\theta) \tag{122}$$

## B.2    PROOF OF THEOREM 4.1

After writing the program using TP operations, we are ready to prove Theorem 4.1 by taking the infinite-width limit. First, note that from Eqs. (119), (120) and (166) and **??** and applying Definition 5.3, we have that:

$$\forall_{1\leq l\leq L}, Z^{\tilde{h}_t^l} = Z^{\tilde{h}^l} \tag{123}$$

$$\forall_{1\leq l\leq L}, Z^{\tilde{x}_t^l} = Z^{\tilde{x}^l} \tag{124}$$

$$\forall_{1\leq l<L}, Z^{d\tilde{h}_t^l} = Z^{d\tilde{h}^l} = Z^{W^{l+1}d\tilde{h}^{l+1}}\phi'(Z^{\tilde{h}^l}) \tag{125}$$

$$Z^{d\tilde{h}_t^L} = Z^{d\tilde{h}^L} = Z^{w^{L+1}}\phi'(Z^{\tilde{h}^L}) \tag{126}$$

Applying Definition 5.3 to Eqs. (117) and (118), we have that:

$$Z^{\delta \tilde{h}^2} = -\mathbb{E}_{Z^{x^1}, Z^{\tilde{x}^1}}\big[Q\big(Z^{dh^2}Z^{x^1}\mathring{\mathcal{L}}'\big)Z^{\tilde{x}^1}\big] \tag{127}$$

$$Z^{\delta \tilde{x}^2} = \phi'(Z^{\tilde{h}^2})Z^{\delta \tilde{h}^2} \tag{128}$$

$$\tag{129}$$

And similarly for Eqs. (113) and (116):

$$\forall_{2<l\leq L} Z^{\delta\tilde{h}^l} = -\mathbb{E}_{Z^{x^{l-1}},Z^{\tilde{x}^{l-1}}}\big[Q\big(Z^{dh^l}Z^{x^{l-1}}\mathring{\mathcal{L}}'\big)Z^{\tilde{x}^{l-1}}\big] + Z^{W^l\delta\tilde{x}^{l-1}} \tag{130}$$

$$\forall_{2\leq l\leq L} Z^{\delta\tilde{x}^l} = \phi'(Z^{\tilde{h}^l})Z^{\delta\tilde{h}^l} \tag{131}$$

$$\tag{132}$$

where we have that $Z^{W^l\delta\tilde{x}^{l-1}} = \hat{Z}^{W^l\delta\tilde{x}^{l-1}} + \dot{Z}^{W^l\delta\tilde{x}^{l-1}}$ according to Definition 5.3. Then using Theorem 5.4:

$$\Delta\tilde{f} = \mathbb{E}\big[Z^{w^{L+1}}Z^{\delta\tilde{x}^L_t}\big] \tag{133}$$

$$= \mathbb{E}\big[Z^{w^{L+1}}\phi'(Z^{\tilde{h}^L})Z^{\delta\tilde{h}^L}\big] \tag{134}$$

$$= -\mathbb{E}\Big[Z^{w^{L+1}}\phi'(Z^{\tilde{h}^L})\mathbb{E}_{Z^{x^{L-1}},Z^{\tilde{x}^{L-1}}}\big[Q\big(Z^{dh^L}Z^{x^{L-1}}\mathring{\mathcal{L}}'_t\big)Z^{\tilde{x}^{L-1}}\big]\Big] \tag{135}$$

$$+ \mathbb{E}\big[Z^{w^{L+1}}\phi'(Z^{\tilde{h}^L})Z^{W^L\delta\tilde{x}^{L-1}}\big] \tag{136}$$

$$= -\mathbb{E}\big[Z^{d\tilde{h}^L}Q\big(Z^{dh^L}Z^{x^{L-1}}\mathring{\mathcal{L}}'_t\big)Z^{\tilde{x}^{L-1}}\big] \tag{137}$$

$$+ \mathbb{E}\big[Z^{d\tilde{h}^L}\dot{Z}^{W^L\delta\tilde{x}^{L-1}}\big] \tag{138}$$

We now use lemma L.3 from Yang (2020b) restated:

**Lemma B.1.** *For any $x, y \in \mathbb{R}^n$ and $W \in \mathbb{R}^{n\times n}$ in the program, it holds that:*

$$\mathbb{E}[Z^x\dot{Z}^{Wy}] = \mathbb{E}[Z^{W^\top x}Z^y] \tag{139}$$

Applying Lemma B.1 to Eq. (138):

$$\mathbb{E}\big[Z^{d\tilde{h}^L}\dot{Z}^{W^L\delta\tilde{x}^{L-1}}\big] = \mathbb{E}\big[Z^{W^{L\top}d\tilde{h}^L}Z^{\delta\tilde{x}^{L-1}}\big] \tag{140}$$

$$= -\mathbb{E}\big[Z^{d\tilde{h}^{L-1}}\big(\mathbb{E}_{Z^{x^{L-2}},Z^{\tilde{x}^{L-2}}}\big[Q\big(Z^{dh^{L-1}}Z^{x^{L-2}}\mathring{\mathcal{L}}'\big)Z^{\tilde{x}^{L-2}}\big] - Z^{W^{L-1}\delta\tilde{x}^{L-2}}\big)\big] \tag{141}$$

$$= -\mathbb{E}\big[Z^{d\tilde{h}^{L-1}}Q\big(Z^{dh^{L-1}}Z^{x^{L-2}}\mathring{\mathcal{L}}'\big)Z^{\tilde{x}^{L-2}}\big] + \mathbb{E}\big[Z^{d\tilde{h}^{L-1}}Z^{W^{L-1}\delta\tilde{x}^{L-2}}\big] \tag{142}$$

Similarly expanding $\mathbb{E}\big[Z^{d\tilde{h}^{L-1}}Z^{W^{L-1}\delta\tilde{x}^{L-2}}\big]$ we arrive at:

$$\Delta\tilde{f} = -\sum_{l=2}^{L}\mathbb{E}\big[Z^{d\tilde{h}^l}Q\big(Z^{dh^l}Z^{x^{l-1}}\mathring{\mathcal{L}}'_t\big)Z^{\tilde{x}^{l-1}}\big] = -\mathcal{K}(\xi,\tilde{\xi}|\mathring{\mathcal{L}}') \tag{143}$$

Finally, using Eq. (152):

$$\Delta\tilde{f}_t = -\mathcal{K}_{\text{adp}}(\xi_t,\tilde{\xi}|\mathring{\mathcal{L}}'_t) \tag{144}$$

### B.3 THE TENSOR PROGRAM FOR THEOREM 4.2

In the next section, we construct the Tensor Program that encodes the training of an 2-hidden layer MLP as in Eq. (1) under the $\mu$ parameterization. Since the last layer is not trained, we define $\bar{w}^3 = \sqrt{n}w^3$, so the output is given by $f(\xi) = \frac{1}{n}\bar{w}^{3\top}x^2(\xi)$. Here we first describe the initial matrices, vectors, and scalars of the program, along with necessary notations.

**Initial Matrices, Vectors and Scalars** We first define the initial set of matrices $\mathcal{W}$, vectors $\mathcal{V}$ and scalars $\mathcal{C}$:

- Initial matrices $w^2 \in \mathbb{R}^{n\times n}$ sampled iid from $\mathcal{N}(0, \frac{1}{n})$. We set $\mathcal{W} = W^2$.

- The initial vectors $\mathcal{V}$ are given by the first layer $h^1(\xi)$ for all inputs, and the last weight vector $\bar{w}^3 \in \mathbb{R}^n$. Notice that $\bar{w}^3$ is normally distributed.

- Initial scalar $\mathcal{C} = \{\frac{1}{\sqrt{n}}\}$.

**Notations**    As in the ANTK case, we use := to more clearly denote assignment happening in the program, as opposed to mathematical equality. To clearly demonstrate the application of TENSOR, we will also freely introduce function symbols $\psi$ to put things into TENSOR form.

**Initial Forward and Backward Passes**    Starting with our initial vectors $h^1(\xi) := w^1\xi$ , we compute $x^1(\xi), h^2(\xi), x^2(\xi), dh^2(\xi), f(\xi), \mathcal{L}'(\xi)$ for all inputs using TENSOR, TENSORMOMENT and MATMUL instructions at step any $t$:

$$x^1(\xi) := \text{TENSOR}(h^1(\xi)) \text{ for } \text{TENSOR}(y) \stackrel{\text{def}}{=} \phi(y) \tag{145}$$

$$h^2(\xi) := W^2 \boldsymbol{x}^1(\xi) \tag{146}$$

$$x^2(\xi) := \text{TENSOR}(h^2(\xi)) \text{ for } \text{TENSOR}(y) \stackrel{\text{def}}{=} \phi(y) \tag{147}$$

$$f(\xi) := \text{TENSORMOMENT}(\bar{w}^3, x^2(\xi)) \text{ for } \text{TENSORMOMENT}(y^1, y^2) \stackrel{\text{def}}{=} \frac{1}{n}\sum_{\alpha=1}^{n} y_\alpha^1 y_\alpha^2 \tag{148}$$

$$\mathcal{L}'(\xi) := \mathcal{L}'_t := \text{TENSORMOMENT}(; f_t) \text{ for } \text{TENSORMOMENT}(; \theta) \stackrel{\text{def}}{=} \frac{1}{n}\sum_{\alpha=1}^{n} \mathcal{L}'(\theta) \tag{149}$$

$$dh^2(\xi) := \text{TENSOR}(\bar{w}^3, h^2(\xi)) \text{for } \text{TENSOR}(y^1, y^2) \stackrel{\text{def}}{=} y^1 \odot \phi'(y^2) \tag{150}$$

Note that with $\mu$ parameterization we can express the output $f(\xi)$ directly without conditioning using a TENSORMOMENT.

**Expressing $\tilde{h}_{t+1}^2$**    Using Eq. (2), we have:

$$\tilde{h}_{t+1}^2 := \text{TENSOR}(\tilde{h}_t^2, dh_t^2, x_t^1, \tilde{x}_t^1; \mathcal{L}'_t) \text{ for } \text{TENSOR}(y^1, y^2, y^3, y^4; \theta) \stackrel{\text{def}}{=} y^1 - \frac{1}{n}Q(y^2 y^{3\top}\theta)y^4 \tag{151}$$

### B.4    PROOF OF THEOREM 4.2

After writing the program using TP operations, we are ready to prove Theorem 4.2 by taking the infinite-width limit. Applying Theorem 5.4 to Eqs. (145) to (151), we have that:

$$Z^{\tilde{h}_{t+1}^2} = Z^{\tilde{h}_t^2} - \mathbb{E}_{Z^{x^1(\xi_t)}, Z^{\tilde{x}^1}}\left[Q(Z^{dh_t^2} Z^{x^1(\xi_t)} \mathring{\mathcal{L}}_t)Z^{\tilde{x}^1}\right] \tag{152}$$

$$Z^{\tilde{x}_t^2} = \phi(Z^{\tilde{h}_t^2}) \tag{153}$$

$$Z^{\tilde{dh}_t^2} = Z^{\tilde{x}_t^1}\phi'(Z^{\tilde{h}_t^2}) \tag{154}$$

$$\tilde{f}_t = \mathbb{E}[Z^{\bar{w}^3} Z^{\tilde{x}_t^2}] \tag{155}$$

where $Z^{\bar{w}^3} \sim \mathcal{N}(0, 1)$.

## C    EXTENSIONS OF THEOREM 4.1 AND THEOREM 4.2 TO AGOS WITH MEMORY

So far we have dealt with the case of memoryless adaptive optimizers, and a batchsize of 1, however our results can be trivially extended to the more general case. To illustrate this, we now show how the proofs of Theorem 4.1 and Theorem 4.2 can be easily adapted to general AGOs with memory, and a general batchsize. Recall from definition Definition 3.1, if $g_0, g_1, ..., g_t \in \mathbb{R}$ denote gradients of some scalar parameter $w$ at times $0, 1, ..., t$, a general adaptive update can be described by a function $Q_t \in \mathbb{R}^{t+1} \to \mathbb{R}$ such that $\Delta w_t \propto Q_t(g_0, g_1, ..., g_t; \epsilon)$. Concretely, in the case of Adam, $Q$ takes the following form (replacing $\beta_1, \beta_2$ with $\gamma_1, \gamma_2$ to prevent confusion with other indices):

$$Q_t(g_0, g_1, ..., g_t; \epsilon) \stackrel{\text{def}}{=} \frac{\frac{(1-\gamma_1)}{1-\gamma_1^t}\sum_{i=0}^{t}\gamma_1^{t-i}g_i}{\sqrt{\frac{(1-\gamma_2)}{1-\gamma_2^t}\sum_{i=0}^{t}\gamma_2^{t-i}g_i^2} + \epsilon} \tag{156}$$

In the context of optimizing an MLP, we can write the equivalent of Eq. (2) for a general $Q$ function, and a general batchsize for both parameterizations:

$$\forall_{1 < l \leq L}, \quad \Delta w_t^l = \tag{157}$$

$$-\frac{1}{n} Q_t \Big( \frac{\sum_{\boldsymbol{\beta}_0} dh_{\boldsymbol{\beta}_0}^l x_{\boldsymbol{\beta}_0}^{l-1\top} \mathcal{L}'_{\boldsymbol{\beta}_0}}{n}, \frac{\sum_{\boldsymbol{\beta}_1} dh_{\boldsymbol{\beta}_1}^l x_{\boldsymbol{\beta}_1}^{l-1\top} \mathcal{L}'_{\boldsymbol{\beta}_1}}{n}, ..., \frac{\sum_{\boldsymbol{\beta}_t} dh_{\boldsymbol{\beta}_t}^l x_{\boldsymbol{\beta}_t}^{l-1\top} \mathcal{L}'_{\boldsymbol{\beta}_t}}{n}; \frac{\epsilon}{n} \Big) \tag{158}$$

$$= -\frac{1}{n} Q_t \Big( \sum_{\boldsymbol{\beta}_0} dh_{\boldsymbol{\beta}_0}^l x_{\boldsymbol{\beta}_0}^{l-1\top} \mathcal{L}'_{\boldsymbol{\beta}_0}, \sum_{\boldsymbol{\beta}_1} dh_{\boldsymbol{\beta}_1}^l x_{\boldsymbol{\beta}_1}^{l-1\top} \mathcal{L}'_{\boldsymbol{\beta}_1}, ..., \sum_{\boldsymbol{\beta}_t} dh_{\boldsymbol{\beta}_t}^l x_{\boldsymbol{\beta}_t}^{l-1\top} \mathcal{L}'_{\boldsymbol{\beta}_t}; \epsilon \Big) \tag{159}$$

where $\sum_{\boldsymbol{\beta}_t}$ denotes summation over samples in the minibatch $\boldsymbol{\beta}_t$ at step $t$ (i.e if $\boldsymbol{\beta}_t := \{\xi_i, \xi_j, \xi_k\}$ then $\sum_{\boldsymbol{\beta}_t} u_{\boldsymbol{\beta}_t} = u_t(\xi_i) + u_t(\xi_j) + u_t(\xi_k)$), and $Q_t$ operates element-wise on the components of its inputs. Note that we have used Definition 3.1 to conveniently remove the $\frac{1}{n}$ factors from inside the $Q$ function, as in Eq. (2). Since $\epsilon$ is a constant, we will absorb it into the definition of $Q$ from now onward. Note that for any vector $v$, the matrix vector product $\Delta w_t^l v$ can be implemented as a TENSOR instruction (see Definition 5.1):

$$\Delta w_t^l v = \text{TENSOR}(\{dh_{\boldsymbol{\beta}_0}^l\}, \{x_{\boldsymbol{\beta}_0}^{l-1}\}, ..., \{dh_{\boldsymbol{\beta}_t}^l\}, \{x_{\boldsymbol{\beta}_t}^{l-1}\}_{\boldsymbol{\beta}_t}, v; \{\mathcal{L}'_{\boldsymbol{\beta}_t}\}) \tag{160}$$

$$:= -\frac{1}{n} Q_t \Big( \sum_{\boldsymbol{\beta}_0} dh_{\boldsymbol{\beta}_0}^l x_{\boldsymbol{\beta}_0}^{l-1\top} \mathcal{L}'_{\boldsymbol{\beta}_0}, ..., \sum_{\boldsymbol{\beta}_t} dh_{\boldsymbol{\beta}_t}^l x_{\boldsymbol{\beta}_t}^{l-1\top} \mathcal{L}'_{\boldsymbol{\beta}_t} \Big) v \tag{161}$$

where $\{u_{\boldsymbol{\beta}_t}\}_{\boldsymbol{\beta}_t}$ is a collection of all vectors $u$ evaluated at time $t$ on minibatch $\boldsymbol{\beta}_t$ (and likewise for scalars). We can now conveniently plug in Eq. (160) into the tensor programs in Theorem 4.1 and Theorem 4.2 to prove a more general result.

## C.1 EXTENSION OF THEOREM 4.1 (NTK)

We now state a general theorem for an AGO with memory and arbitrary batchsize.

**Theorem C.1.** *Let $f(\xi) \in \mathbb{R}$ denote an MLP as in Eq. (1) parameterized using the ANTK parameterization described in Section 4.1, where $\phi'$ is pseudo-Lipschitz. Assume layers $\{w^l\}_{l=2}^L$ are trained using an AGO applied on minibatches of arbitrary size, where $Q_t$ is pseudo-Lipschitz defined according to Definition 3.1, using a loss function $\mathcal{L}$ with a pseudo-Lipschitz first derivative. Then, at any step $t$ and for any sample $\tilde{\xi}$, it holds that $\tilde{f}_t \overset{a.s}{\to} \mathring{\tilde{f}}_t$ where $\Delta \mathring{\tilde{f}}_t = -\mathcal{K}_{Adp}(\{\xi_{\boldsymbol{\beta}_0}\}, ..., \{\xi_{\boldsymbol{\beta}_t}\}, \tilde{\xi} | \{\mathring{\mathcal{L}}'_{\boldsymbol{\beta}_0}\}, ..., \{\mathring{\mathcal{L}}'_{\boldsymbol{\beta}_t}\})$, where:*

$$\mathcal{K}_{Adp}(\{\xi_{\boldsymbol{\beta}_0}\}, ..., \{\xi_{\boldsymbol{\beta}_t}\}, \tilde{\xi} | \{\mathring{\mathcal{L}}'_{\boldsymbol{\beta}_0}\}, ..., \{\mathring{\mathcal{L}}'_{\boldsymbol{\beta}_t}\}) = \tag{162}$$

$$\sum_{l=2}^L \mathbb{E}\Big[ Z^{d\tilde{h}^l} Q_t \Big( \sum_{\boldsymbol{\beta}_0} Z^{dh_{\boldsymbol{\beta}_0}^l} Z^{x_{\boldsymbol{\beta}_0}^{l-1}} \mathring{\mathcal{L}}'_{\boldsymbol{\beta}_0}, ..., \sum_{\boldsymbol{\beta}_t} Z^{dh_{\boldsymbol{\beta}_t}^l} Z^{x_{\boldsymbol{\beta}_t}^{l-1}} \mathring{\mathcal{L}}'_{\boldsymbol{\beta}_t} \Big) Z^{\tilde{x}^{l-1}} \Big] \tag{163}$$

$$\mathring{\mathcal{L}}'_t = \mathcal{L}'_t(\mathring{f}_t(\xi_t)) \tag{164}$$

*where the expectation is taken over all $Z$ variables at initialization.*

*Proof.* The proof of Theorem C.1 is a straightforward extension of the proof of Theorem 4.1. The forward and backward passes for any $t$ are again given by:

$$\tilde{h}_t^l = \Big( W^l + \frac{1}{\sqrt{n}} \sum_{t'=0}^{t-1} \Delta w_{t'}^l \Big) \tilde{x}_t^{l-1} \tag{165}$$

$$d\tilde{h}_t^L = \phi'(\tilde{h}_t^L) \odot w^{L+1} \tag{166}$$

$$\forall_{1 \leq l < L}, \ d\tilde{h}_t^l = \Big[ \Big( W^{l+1} + \frac{1}{\sqrt{n}} \sum_{t'=0}^{t-1} \Delta w_{t'}^{l+1} \Big)^\top d\tilde{h}_t^{l+1} \Big] \odot \phi'(\tilde{h}_t^l) \tag{167}$$

only with weight updates that are given by:

$$\Delta w_t^l = -\frac{1}{n} Q_t \Big( \sum_{\boldsymbol{\beta}_0} dh_{\boldsymbol{\beta}_0}^l x_{\boldsymbol{\beta}_0}^{l-1\top} \mathcal{L}'_{\boldsymbol{\beta}_0}, ..., \sum_{\boldsymbol{\beta}_t} dh_{\boldsymbol{\beta}_t}^l x_{\boldsymbol{\beta}_t}^{l-1\top} \mathcal{L}'_{\boldsymbol{\beta}_t} \Big) \tag{168}$$

As in the memoryless case, using Eq. (2) and $w_t^l = w^l + \sum_{t'=0}^{t} \Delta w_{t'}^l$ (recall $w^l = \sqrt{n}W^l$), we have that:

$$\delta \tilde{h}_t^2 = \Delta w_t^2 \tilde{h}^1 \tag{169}$$

$$\forall_{2 < l \le L}, \quad \delta \tilde{h}_t^l = \Delta w_t^l \tilde{x}_t^{l-1} + \frac{1}{\sqrt{n}}\left(w^l + \sum_{t'=0}^{t} \Delta w_{t'}^l\right)\delta \tilde{x}_t^{l-1} + \frac{1}{\sqrt{n}}\Delta w_t^l \delta \tilde{x}_t^{l-1} \tag{170}$$

$$\delta \tilde{x}_t^l = \sqrt{n}\phi\left(\tilde{h}_t^l + \frac{\delta \tilde{h}_t^l}{\sqrt{n}}\right) - \sqrt{n}\phi(\tilde{h}_t^l) \tag{171}$$

For any vector $v$, the matrix vector product $\Delta w_t^l v$ can be implemented as a TENSOR instruction:

$$\Delta w_t^l v = \text{TENSOR}(\{dh_{\boldsymbol{\beta}_0}^l\}, \{x_{\boldsymbol{\beta}_0}^{l-1}\}, ..., \{dh_{\boldsymbol{\beta}_t}^l\}, \{x_{\boldsymbol{\beta}_t}^{l-1}\}_{\boldsymbol{\beta}_t}, v; \{\mathcal{L}'_{\boldsymbol{\beta}_t}\}) \tag{172}$$

$$= -\frac{1}{n}Q_t\left(\sum_{\boldsymbol{\beta}_0} dh_{\boldsymbol{\beta}_0}^l x_{\boldsymbol{\beta}_0}^{l-1\top}\mathcal{L}'_{\boldsymbol{\beta}_0}, ..., \sum_{\boldsymbol{\beta}_t} dh_{\boldsymbol{\beta}_t}^l x_{\boldsymbol{\beta}_t}^{l-1\top}\mathcal{L}'_{\boldsymbol{\beta}_t}\right)v \tag{173}$$

where $\{u_{\boldsymbol{\beta}_t}\}_{\boldsymbol{\beta}_t}$ is a collection of all vectors $u$ evaluated at time $t$ on minibatch $\boldsymbol{\beta}_t$ (and likewise for scalars). Hence, we may proceed exactly as in the base proof of Theorem 4.1 (i.e expressing the optimization process as a tensor program, applying Definition 5.3 to get the coordinate distributions in the limit, and applying Theorem 5.4). Note that in the concrete case of Adam, we get the following function update:

$$\mathcal{K}_{\text{Adp}}(\{\xi_{\boldsymbol{\beta}_0}\}, ..., \{\xi_{\boldsymbol{\beta}_t}\}, \tilde{\xi}|\{\mathring{\mathcal{L}}'_{\boldsymbol{\beta}_0}\}, ..., \{\mathring{\mathcal{L}}'_{\boldsymbol{\beta}_t}\}) = \tag{174}$$

$$\sum_{l=2}^{L}\mathbb{E}\left[Z^{d\tilde{h}^l}\frac{\frac{(1-\gamma_1)}{1-\gamma_1^t}\sum_{i=0}^{t}\gamma_1^{t-i}\sum_{\boldsymbol{\beta}_i}Z^{dh_{\boldsymbol{\beta}_i}^l}Z^{x_{\boldsymbol{\beta}_i}^{l-1}}\mathring{\mathcal{L}}'_{\boldsymbol{\beta}_i}}{\sqrt{\frac{(1-\gamma_2)}{1-\gamma_2^t}\sum_{i=0}^{t}\gamma_2^{t-i}(\sum_{\boldsymbol{\beta}_i}Z^{dh_{\boldsymbol{\beta}_i}^l}Z^{x_{\boldsymbol{\beta}_i}^{l-1}}\mathring{\mathcal{L}}'_{\boldsymbol{\beta}_i})^2} + \epsilon}Z^{\tilde{x}^{l-1}}\right] \tag{175}$$

$\square$

## C.2 EXTENSION OF THEOREM 4.2 ($\mu$P)

**Theorem C.2.** *Let $f(\xi) \in \mathbb{R}$ denote an MLP as in Eq. (1) with $L = 2$ parameterized using the $\mu$ parameterization described in Section 4.1, where $\phi'$ is pseudo-Lipschitz. Assume layers $w^2$ is trained using an AGO with a pseudo-Lipschitz function $Q$ function according to Definition 3.1 (for general batchsize), using a loss function $\mathcal{L}$ with a pseudo-Lipschitz first derivative. Then at any step $t$ and for any sample $\tilde{\xi}$, it holds that $\tilde{f}_t \xrightarrow{a.s} \mathring{\tilde{f}}_t$ where $\mathring{\tilde{f}}_t$ can be computed as follows:*

$$Z^{\tilde{h}_{t+1}^2} = Z^{\tilde{h}_t^2} - \mathbb{E}_{Z^{x^1(\xi_\bullet)}, Z^{\tilde{x}^1}}\left[Q_t\left(\sum_{\boldsymbol{\beta}_0}\zeta\phi'(Z^{h_{\boldsymbol{\beta}_0}^2})Z^{x^1(\xi_{\boldsymbol{\beta}_0})}\mathring{\mathcal{L}}_{\boldsymbol{\beta}_0}, ..., \sum_{\boldsymbol{\beta}_t}\zeta\phi'(Z^{h_{\boldsymbol{\beta}_0}^2})Z^{x^1(\xi_{\boldsymbol{\beta}_t})}\mathring{\mathcal{L}}_{\boldsymbol{\beta}_t}\right)Z^{\tilde{x}^1}\right] \tag{176}$$

$$Z^{\tilde{x}_t^2} = \phi(Z^{\tilde{h}_t^2}), \quad \mathring{\tilde{f}}_0 = 0, \quad \tilde{f}_t = \mathbb{E}[\zeta Z^{\tilde{x}_t^2}], \quad \mathring{\mathcal{L}}'_t = \mathcal{L}'_t(\mathring{f}_t(\xi_t)) \tag{177}$$

*where the expectations are taken over all $Z$ variables (including $\zeta \stackrel{\mathrm{d}}{=} \mathcal{N}(0, 1)$).*[4]

*Proof.* A similarly straightforward application of the Master Theorem Theorem 5.4 to the tensor program in described in Theorem 4.2 together with Eq. (160) in $\mu$P. $\square$

## D NUMERICAL VERIFICATION

We conduct numerical experiments to verify our results. For both parameterizations, the exact network dynamics at the infinite width limit is not tractable in the general case, since the expectations involved do not admit an analytical solution (unlike the standard NTK for ReLU networks). Even for

---

[4]Once again, the loss derivatives $\mathcal{L}'_t$ are deterministic in Eq. (8)

the ANTK parameterization, the infinite-width dynamics cannot be separated to a fixed kernel and a loss derivative, as with the NTK dynamics for SGD. We therefore must resort to MC simulations to approximate the expectations involved in evaluating the infinite width dynamics in both regimes. We verify Theorem C.1 and Theorem C.2 by training a ReLU MLP ($L = 4$ for ANTK and $L = 2$ for $\mu$) on $\mathbb{R}^{10}$ gasussian inputs and a unit output. For a loss we use the standard L2 loss function, regressing to random targets. We train networks with varying widths using Adam with $\beta_1 = 0.9, \beta_2 = 0.99$ in full batch mode, on 100 training samples, and run 10 trials per width. We use a learning rate of $\frac{0.2}{n}$, and $\epsilon = \frac{1e-4}{n}$ (where $n$ is the width). In order to account for different initial outputs and loss derivative per weight initialization, we subtract the initialized network output from the output for each sample, such that the output is identically zero at initialization for all inputs. To approximate the infinite-width training dynamics, we approximate the expectation in Eq. (174) and Eq. (176) using MC simulations where we sample the $Z$ random variables from gaussian processes corresponding to the network architecture at initialization. Since the initial loss derivatives are deterministic (given that the outputs are zero), the infinite width dynamics can be approximated without actually constructing a network. To compare the evolution of the finite vs infinite architectures, we evaluate the output at each iteration on random inputs. Our results are summarized in Fig. 2 and Fig. 3. As expected, as the width increases the training dynamics converge to that of the infinite dynamics.

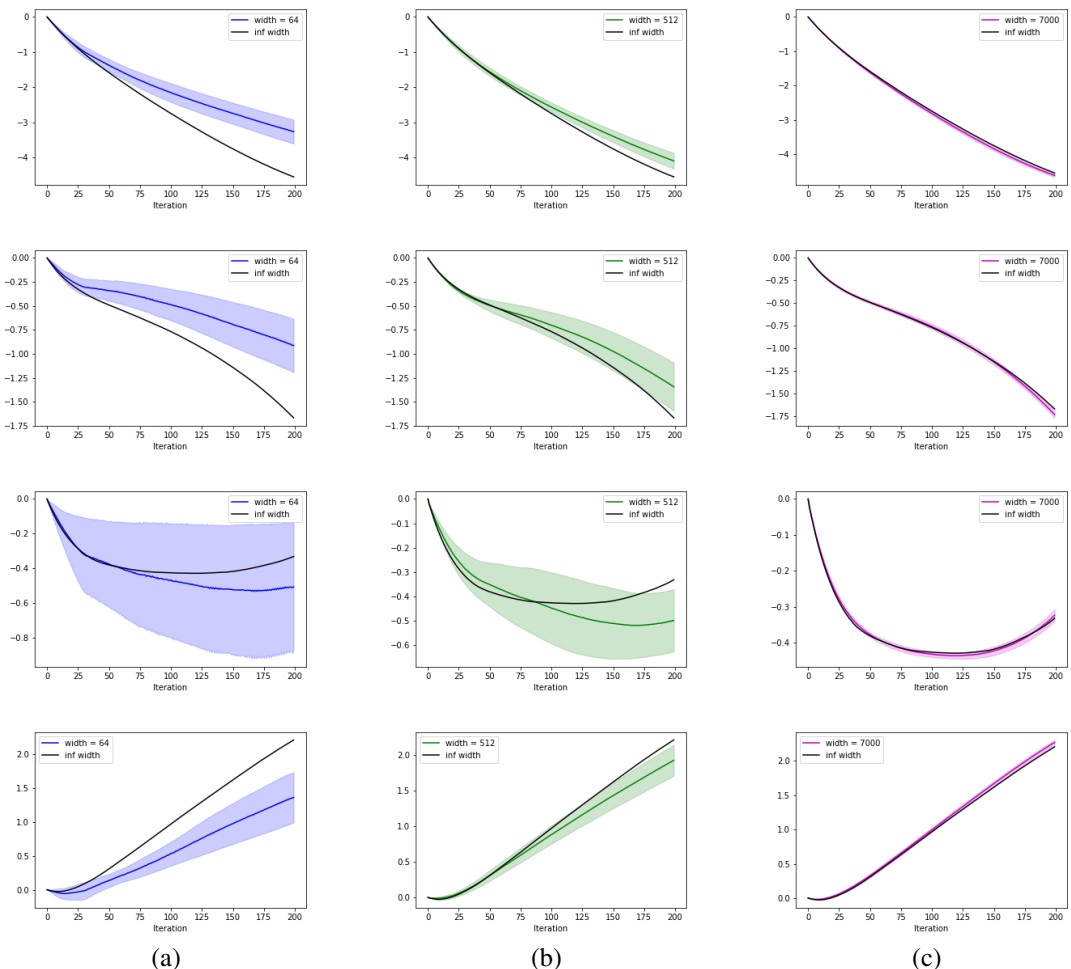

(a)     (b)     (c)

Figure 2: Training dynamics of finite and infinite-width networks in the ANTK parameterization. We train networks of widths 64 (a), 512 (b), 7000 (c) , and track the outputs for 4 random inputs (one per row) at each iteration as the network trains. We compute the output distribution over 10 independent runs for each network, and compare with the infinite-width dynamics (black curve). As the width grows, the network function converges to that of the infinite-width dynamics captured in Eq. (174)

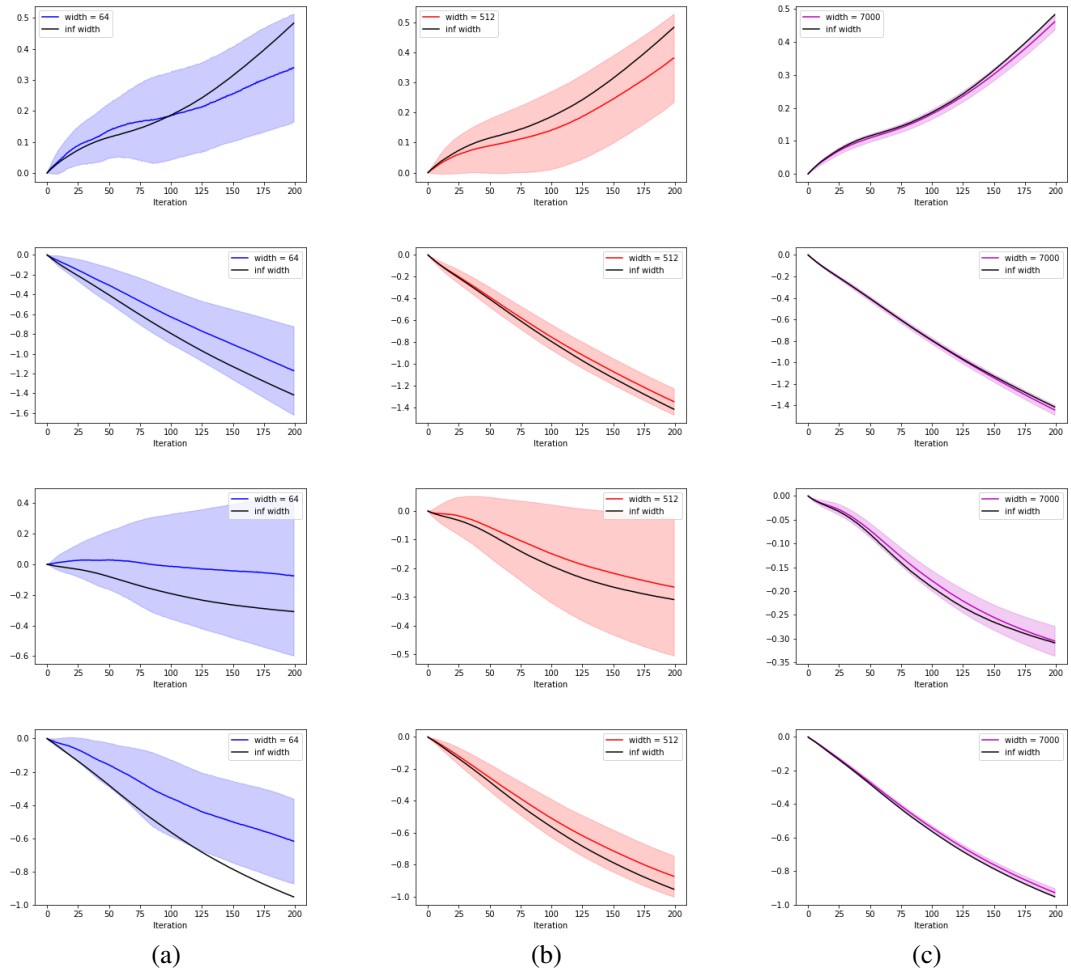

Figure 3: Training dynamics of finite and infinite-width networks in the $\mu$ parameterization. We train networks of widths 64 (a), 512 (b), 7000 (c) , and track the outputs for 4 random inputs (one per row) at each iteration as the network trains. We compute the output distribution over 10 independent runs for each network, and compare with the infinite-width dynamics (black curve). As the width grows, the network function converges to that of the infinite-width dynamics captured in Eq. (176)

