# OpenReview forum: "Adaptive Optimization in the $\infty$-Width Limit"
_ICLR.cc/2023/Conference — ICLR 2023 poster_

### Official Review · Reviewer_8D7G · 2022-10-24

**Confidence:** 3
**Correctness:** 3
**Technical Novelty And Significance:** 3
**Empirical Novelty And Significance:** 3
**Recommendation:** 6

**Clarity, Quality, Novelty And Reproducibility:**

The paper is clear and contributions are novel.


**Strength And Weaknesses:**

Strengths:

- Infinite width limit of MLP studied for the first time with adaptive optimizers.


Weaknesses:

- No empirical data is provided in support of theoretical findings.

- NTK limit is provided (theorem 4.1) in memoryless case only, which is of very limited interest.
The authors state: "we maintain that our analysis is applicable in the more general setting without introducing any additional arguments", but no proof is provided as far as I understand.

(It has been addressed in the rebuttal)


**Summary Of The Paper:**

This paper studies the infinite width limit of Multi-layer perceptrons, in particular the existence of feature learning.
In contrast to previous work, it studies adaptive optimizers instead of Gradient Descent, but results are roughly the same: NTK limit still results in lazy training (and no feature learning), while mu-parameterization supports feature learning.
Unfortunately, no experiments on real (or even synthetic) data are provided in support of theoretical claims (It has been addressed in the rebuttal).


**Summary Of The Review:**

I would be much more convinced of the significance of this work if empirical validation of the theoretical results was provided (It has been addressed in the rebuttal).

---

> ### Author Response · Authors · 2022-11-09
> **Rebuttal**
>
> Thanks for your time!
>
> > No empirical data is provided in support of theoretical findings.
>
> We have added experiments in the new draft (section D, appendix)
>
> > NTK limit is provided (theorem 4.1) in memoryless case only, which is of very limited interest. The authors state: "we maintain that our analysis is applicable in the more general setting without introducing any additional arguments", but no proof is provided as far as I understand.
>
> We have stated the limit equations for Adam, with general batch size, in the new draft (section C in the appendix).
>
> If our response satisfies the reviewer, please consider raising your score. Thank you!

---

> > ### Comment · Reviewer_8D7G · 2022-12-04
> > **Answer to rubuttal**
> >
> > Thank you for providing Appendix C and D, I'm raising the score.
> >
> > Could you please correctly refer to all appendices in the main text.
> > Currently the only reference is to Appendix C, stating incorrectly (both in main text and in "Appendix organization") that it proves Theorems 4.1 and 4.2. Instead, Theorems 4.1 and 4.2 are proven in Appendix B, not C
> >
> > Furthermore, Appendix C should be referenced in the main text, ideally multiple times.
> > While the authors seem to think that the results in Appendix C are trivial, *all four* reviewers asked to include that case, so the authors should make it very clear to readers in the main text, even if the formal proof is provided in Appendix C.
> > A similar comment applies to Appendix D, it should be referenced to in the main text (and in "Appendix organization").
> >
> > Minor:
> > Please provide y axis labels in Fig.2

---

> > > ### Author Response · Authors · 2022-12-09
> > > **Authors response**
> > >
> > > Thank you for your feedback and support. We apologize for any confusion regarding the references to Appendix C and D. We will correct the error in the main text and in the "Appendix Organization", and add references to the relevant appendices in the main text as suggested, as well as provide y-axis labels in Fig. 2.

---

### Official Review · Reviewer_k9hR · 2022-10-24

**Confidence:** 3
**Correctness:** 3
**Technical Novelty And Significance:** 3
**Empirical Novelty And Significance:** Not applicable
**Recommendation:** 8

**Clarity, Quality, Novelty And Reproducibility:**

Sections 1-4 are very clear and easy to read. Sections 5 and 6 are more difficult; however, I believe that this is due to the technical nature of the content, and not any fault of the authors who make an effort to build intuition for the proofs. The arguments are, to my knowledge, novel.

**Strength And Weaknesses:**

The arguments for the memoryless case are very clean. In addition, these additions to the TP framework make it more flexible and can give insight in many of the more complex practical cases which arise when training large models. The authors make a good mix of intuition-building arguments and links to the more rigorous ones.

One point of clarification: in the mean-field limit, does $\epsilon$ have to go to $0$ to get all the scaling arguments right? Is there an intuitive reason that this doesn't mess up the Gaussianity arguments via divisors which are random variables which can have significant probability mass around $0$? The final limits seem to be $\epsilon$-independent.

One of the big weaknesses is that the promise of extending the analysis to the case of an adaptive optimizer with memory (depending on gradients $g$ further back in time) is not clearly made. I believe without that, the result in the memoryless case is interesting but not sufficiently significant for an acceptance rating. It would also be good to have a discussion on how the total time $t$ must scale with $n$ for the arguments to hold.

I think the paper would be significantly strengthened if there was an explicit formulation and proof of at least the NTK for the ADAM algorithm. From the comments in the paper these seem like results that are already completed, or easily derivable for the framework. This paper is positioned to become a standard reference for NTK/$\mu$-parameterization for adaptive algorithms, and it would be strange for the most popular adaptive algorithm to not be explicitly covered.

Due to the limits of my technical ability and the review timescale, I have not been able to rigorously check the main proofs in the TP framework. I followed along with the arguments in the main text and they seem sound. I believe that there needs to be numerical evidence that the proposed scaling limits hold - not to replace the proofs, but to supplement them and decrease the possibility that some key point in the complex machinery did not go awry.

As a small note, on page 3 it is unclear why the labeling of the gradients $g$ has even time indices only.

Update: most of the concerns above have been addressed; my updated review score reflects those changes.

**Summary Of The Paper:**

In this work the authors tackle the question of the infinite-width limit of neural networks trained using adaptive optimizers (like ADAM). They show that if the step taken in the optimizer depends on a non-linear, scale-invariant function of the previous step, then the NTK can be computed by applying said function to $\frac{\partial f_{train}}{\partial\theta}\nabla_{\theta}\mathcal{L}_{train} $, and then multiplying by $\frac{\partial f_{out}}{\partial\theta}$. An analogous result holds for the $\mu$-parameterization (the unique family of infinite-width feature learning initializations).

**Summary Of The Review:**

The paper does an excellent job of building intuitive and rigorous arguments for memoryless adaptive optimizers. However, I feel that this is not a significant enough contribution on its own, and the case of non-memoryless optimizers like ADAM should be explicitly covered if they are in fact trivial extensions of the framework as suggested by the authors. I did not have the time nor expertise to rigorously check the proofs. I also suggest that numerical evidence of convergence to the theoretical limits would be very helpful to readers, especially those without strong theoretical backgrounds.

---

> ### Author Response · Authors · 2022-11-09
> **Rebuttal**
>
> Thank you for your time!
>
> > One point of clarification: in the mean-field limit, does ϵ have to go to 0 to get all the scaling arguments right? Is there an intuitive reason that this doesn't mess up the Gaussianity arguments via divisors which are random variables which can have significant probability mass around 0? The final limits seem to be ϵ-independent.
>
> The reason eps has to go to 0 is very simple actually: it's the right scale for eps to actually have a nontrivial effect. This is similar to how the learning in muP apparently is scaled to go to 0 but that actually is the right scaling for the learning dynamics to not blow up or get stuck at initialization.
>
> More precisely, the reasoning for the scale of eps is as follows: The adaptive gradient is calculated as m/(sqrt{v}+eps). But sqrt{v} itself has scale 1/n in muP (or 1/sqrt(n) in NTK parametrization). Therefore, if eps is Theta(n^-a) for a < 1, then sqrt{v}+eps ~ eps and v is ignored; if a > 1, then sqrt{v} + eps ~ sqrt{v}, and eps is ignored. Only when a = 1 does eps have nontrivial effect on the denominator. (Similar reasoning for NTK param with a's threshold at 1/2).
>
> > One of the big weaknesses is that the promise of extending the analysis to the case of an adaptive optimizer with memory (depending on gradients g further back in time) is not clearly made.
>
> We have stated the limit equations for Adam, with general batch size, in the new draft (section C in the appendix).
>
> > It would also be good to have a discussion on how the total time t must scale with n for the arguments to hold.
>
> For general Tensor Programs, we expect the limit to hold whenever the width n is exponential in the program length; there are also counterexamples to show the limit do not hold for certain programs breaking this condition. Applied to the program at hand, whose length (morally, the total number of matmuls computed) is linear in t, we see that if the width is exponential in t then we expect to see the limit hold. However, it is still possible for smaller width to work as well, given sufficient assumptions on the structure of the network and/or training data.
>
> > I believe that there needs to be numerical evidence that the proposed scaling limits hold - not to replace the proofs, but to supplement them and decrease the possibility that some key point in the complex machinery did not go awry.
>
> We have added experiments in the new draft (section D, appendix)
>
> > As a small note, on page 3 it is unclear why the labeling of the gradients g has even time indices only.
>
> This is a typo. Thanks for bringing this up!
>
> If our response satisfies the reviewer, please consider raising your score. Thank you!

---

> > ### Comment · Reviewer_k9hR · 2022-11-11
> > **Followup questions**
> >
> > Thanks so much for the quick responses!
> >
> > The point about the scaling of $\epsilon$ is well taken. Is there any comment to be made here that constant $\epsilon$ may lead to bad behavior in practical settings, say, ADAM, with standard parameterization? That fact alone may be interesting to practitioners as they scale up to larger and larger models.
> >
> > Thank you for adding the description of ADAM in this framework. Are there any simple cases where the expectation can be evaluated, to give some intuition to the form of the resulting dynamical equations?
> >
> > Regarding the relationship between width and time: thank you for clarifying. For readers unfamiliar with TP, I think it would be useful to mention this point somewhere in the main text. Maybe when you link to the numerical results, which seem to show that the theory may be useful outside this limit? I believe this paper should attract readers interested in ANTK but not yet with TP, so making some of these points more explicit can help.
> >
> > There are some typos in the updated sections (e.g. "ANTK parameyterization"). In addition, in section 4, "notations" (boldface, start of paragraph) should be capitalized and followed by a colon.
> >
> > Overall the authors have very nicely addressed many of my concerns; I will update my review score accordingly.

---

> > > ### Author Response · Authors · 2022-11-14
> > > **Response to followup questions**
> > >
> > > We appreciate the support! Our response to the reviewers followup questions bellow:
> > >
> > > Q: "Is there any comment to be made here that constant eps may lead to bad behavior in practical settings..."
> > >
> > > A: Yes there is. Even for standard parameterization eps needs to have a scale of \Theta(n^-0.5) to have a nontrivial effect as the width increases. We believe this is indeed relevant for practitioners, however this is not the emphasis of the present paper.
> > >
> > > Q: "Are there any simple cases where the expectation can be evaluated, to give some intuition to the form of the resulting dynamical equations?"
> > >
> > > A: We do not know of any simple cases where Adams dynamical equations have a simple analytical solution, though intuition can still be derived from the form of the dynamical equations. This however is not straight forward and requires substantial additional work, and is outside the scope of the present paper.
> > >
> > > Q: "Regarding the relationship between width and time..."
> > >
> > > A: Point taken. We will mention this issue in the next revision.
> > >
> > > Regarding the various typos, thank you for pointing them out. We will upload a revised draft.

---

### Official Review · Reviewer_1g2Y · 2022-10-25

**Confidence:** 3
**Correctness:** 4
**Technical Novelty And Significance:** 3
**Empirical Novelty And Significance:** 3
**Recommendation:** 5

**Clarity, Quality, Novelty And Reproducibility:**

The writing quality of this paper is quite poor and very informal. It should be improved by providing more comprehensive and rigorous flows.


**Strength And Weaknesses:**

Strength:

- This paper extends two infinite-width limits of MLPs for adaptive gradient-based optimization, which have not been studied before. The proposed results can generalize both kernel and feature learning limits of non-adaptive settings. It is interesting that the adaptive NTK regime is still data-independent like the non-adaptive NTK.

- They provide a general Tensor Program framework for the adaptive optimization setting with O(1/sqrt(n)) convergence rate guarantees.

Weakness:

- Adaptive optimizations have various benefits over the SGD (e.g., fast convergence, better generalization, etc). Can this point of view be captured by the proposed adaptive infinite-width limits? It would be great if more clear motivations and meaningful results of adaptive infinite-width limits are provided.

- The paper focuses on simple settings (e.g., memoryless adaptive gradient descents, update of parameters in a single layer). But, it is unclear whether the proposed results hold for more general settings.


**Summary Of The Paper:**

This paper studies two infinite-width limits (kernel and feature learning limits) of fully-connected neural networks (MLPs) under adaptive gradient-based optimization. Both results generalize that of MLPs trained by a non-adaptive way. In addition, the authors modify a framework (called Tensor Program) that allows to express the adaptive gradient processing as well as the convergence guarantee.


**Summary Of The Review:**

This paper extends infinite-width limits to the adaptive optimization setting, which has never been discovered before. But, it seems that a more in-depth comparison/analysis of the non-adaptive results would make the contributions much stronger. The writing quality needs to be improved for better readability.

---

> ### Author Response · Authors · 2022-11-09
> **Rebuttal**
>
> Thank you so much for your time!
>
> > The paper focuses on simple settings (e.g., memoryless adaptive gradient descents, update of parameters in a single layer). But, it is unclear whether the proposed results hold for more general settings.
>
> We have stated the limit equations for Adam, with general batch size, in the new draft (section C in the appendix).
>
> > The writing quality of this paper is quite poor and very informal. It should be improved by providing more comprehensive and rigorous flows.
>
> This contradicts with reviewer k9hR. We would love to hear your opinion again after your discussion amongst yourselves. In any case, we have been and will be continuing to improve the presentation of this work until it is camera-ready. We would love to hear if you have any concrete suggestions for improving the presentation.
>
> > Adaptive optimizations have various benefits over the SGD (e.g., fast convergence, better generalization, etc). Can this point of view be captured by the proposed adaptive infinite-width limits?
>
> Indeed, we can say more using the infinite-width equations derived here to derive properties of adaptive optimizers. As an elementary example, the adaptive NTK equation can be linearized and solved explicitly. But more advanced analysis is possible with the adaptive NTK and the muP limits.
>
> However, the calculations involved in doing so is quite nontrivial and of a different nature than the mathematics in this paper that lays the foundation. They will themselves in addition take up a lot of space. If we were to bundle all of this here, it would make the narrative much less focused and the derivation confusing, given the length of the paper as it is and the page limit. Thus, as much as the purpose of a paper is to promote understanding (to readers both in the near term and in the long term), it is much better for us to write them as separate papers, where this paper is intended as the foundation for all that would follow.
>
> While we try to make the fruits of our labor seem as intuitive as possible to the reader (which is the duty of any author), it should also be clear from the proofs that this extension of Tensor Programs is highly nontrivial.
>
> Without this foundation, the punchline the reviewer is looking for will never come. As such, we respectively disagree with this criticism.
>
>
> ----------
>
>
> If our response satisfies the reviewer, please consider raising your score. Thank you!

---

### Official Review · Reviewer_7U4M · 2022-10-25

**Confidence:** 2
**Correctness:** 3
**Technical Novelty And Significance:** 3
**Empirical Novelty And Significance:** Not applicable
**Recommendation:** 8

**Clarity, Quality, Novelty And Reproducibility:**

The submission extended the Tensor Programs framework to allow the expression of the computation graph involving adaptive optimizers. From this perspective, the novelty is limited.

**Strength And Weaknesses:**

#### Strength
- Close the gap between adaptive optimizer and NTK analysis.
- Extend the guarantee in NTK and feature learning to the non-linear cases in the sense of the Tensor program language.


#### Weaknesses
- The adaptive optimizer considered in the analysis is memoryless and one-dimensional (Thm.4.1-4.2) given batch size is one, which is far from the practical case.
- The theoretical results seem to be a generalized version of SGD, and we cannot see the difference between AGO and SGD from the theory. In fact, AGO and SGD behaviors differently during training, and we cannot treat SGD as a particular case.

**Summary Of The Paper:**

This submission analyzes the dynamics of adaptive optimization of arbitrary neural network architectures in the infinite-width limit, and as a by-product,  a new tensor program framework is proposed.

**Summary Of The Review:**

The submission adds some new tools in the previously developed framework (Tensor programs) for helping analyze the dynamic of $\infty$-width neural network trained by adaptive optimizers. The only disadvantage is that the theory does not give more insight into the difference between SGD and adaptive optimizers (e.g., Adam).

Overall, I still hold a positive score for this work since it is a necessary expansion of Tensor programs.

---

> ### Author Response · Authors · 2022-11-09
> **Rebuttal**
>
> Thank you so much for your time!
>
> > The adaptive optimizer considered in the analysis is memoryless and one-dimensional (Thm.4.1-4.2) given batch size is one, which is far from the practical case.
>
> We have stated the limit equations for Adam, with general batch size, in the new draft (section C in the appendix).
>
> > The theoretical results seem to be a generalized version of SGD, and we cannot see the difference between AGO and SGD from the theory. In fact, AGO and SGD behaviors differently during training, and we cannot treat SGD as a particular case.
>
> Indeed, we can say more using the infinite-width equations derived here to derive properties of adaptive optimizers. As an elementary example, the adaptive NTK equation can be linearized and solved explicitly. But more advanced analysis is possible with the adaptive NTK and the muP limits.
>
> However, the calculations involved in doing so is quite nontrivial and of a different nature than the mathematics in this paper that lays the foundation. They will themselves in addition take up a lot of space. If we were to bundle all of this here, it would make the narrative much less focused and the derivation confusing, given the length of the paper as it is and the page limit. Thus, as much as the purpose of a paper is to promote understanding (to readers both in the near term and in the long term), it is much better for us to write them as separate papers, where this paper is intended as the foundation for all that would follow.
>
> While we try to make the fruits of our labor seem as intuitive as possible to the reader (which is the duty of any author), it should also be clear from the proofs that this extension of Tensor Programs is highly nontrivial.
>
> Without this foundation, the punchline the reviewer is looking for will never come. As such, we respectively disagree with this criticism.
>
> ----------
>
> If our response satisfies the reviewer, please consider raising your score. Thank you!

---

### Author Response · Authors · 2022-11-11
**Summary of Author Response**

We thank the reviewers for their reviews, which have helped us improve the paper. All reviewers agree on the technical significance and novelty of the paper, however we have updated the draft addressing the following concerns shared by the reviewers:

1) We have added section c (appendix) where we state and prove theorems c.1,c.2, which generalize theorems 4.1 and 4.2 to adaptive optimizers with memory and an arbitrary batchsize. This is in response to a critique common to all reviews on the generalizability of our results to more practical settings. We again would like to emphasize that the additional complexity of proving theorems c.1,c.2 is essentially non existent given theorem 5.4 (the Tensor Program master theorem) beyond introducing additional indexing notations, which would have considerably cluttered the presentation of the main paper.

2) We have added section d (appendix) where we numerically verify our results (specifically theorems c.1, c.2). This is again in response to the reviewers concerns with regard to a lack of numerical verifications.

3) We have fixed various small typos throughout the paper.

We would be happy to address any additional concerns as they arise.

---

### Decision · Program_Chairs · 2023-01-20

**Decision:**

Accept: poster

**Justification For Why Not Higher Score:**

Although the overall assessment is favorable, there were no advocates for a higher score.

**Justification For Why Not Lower Score:**

The reviewers agree that the paper should be accepted.

**Metareview: Summary, Strengths And Weaknesses:**

The paper investigates adaptive optimization in the infinite width limit of neural networks.

* Strengths are a novel and relevant extension of tensor program theory along with analysis of adaptive optimizers advancing the connection of theory and practical settings.
* Weaknesses are certain limitations of the method to discern between training methods.

Reviewers mostly agree on the novelty and technical significance. In the rebuttal the authors addressed some of the reviewers' initial comments. Particularly, generalizations of the results to optimizers with memory and numerical evaluations. The final recommendations were mostly on the favorable side. I find that the topic and methods are of interest and conclude that the paper should be accepted. The authors should still carefully check updates conducted during the rebuttal period, particularly on memory and dimension.


**Note From Pc:**

if the above contains the word "oral" or "spotlight" please see: "oral" presentation means -> notable-top-5% and "spotlight" means -> notable-top-25%. As stated in our emails, we are disassociating presentation type from AC recommendations

**Summary Of Ac-Reviewer Meeting:**

The meeting was called off due to cancellations, with discussions in the forum instead outlined above.